# SorLA restricts TNFα release from microglia to shape a glioma-supportive brain microenvironment

Paulina Kaminska [1,2], Peter L Ovesen[3], Mateusz Jakiel[1,4], Tomasz Obrebski[1], Vanessa Schmidt[3], Michal Draminski[4], Aleksandra G Bilska [1,5], Magdalena Bieniek[2], Jasper Anink[6], Bohdan Paterczyk [1], Anne Mette Gissel Jensen[7], Sylwia Piatek [1], Olav M Andersen[7], Eleonora Aronica[6,8], Thomas E Willnow [3,7], Bozena Kaminska [2], Michal J Dabrowski [4] & Anna R Malik [1,2]✉

## Abstract

SorLA, encoded by the gene *SORL1*, is an intracellular sorting receptor of the VPS10P domain receptor gene family. Although SorLA is best recognized for its ability to shuttle target proteins between intracellular compartments in neurons, recent data suggest that also its microglial expression can be of high relevance for the pathogenesis of brain diseases, including glioblastoma (GBM). Here, we interrogated the impact of SorLA on the functional properties of glioma-associated microglia and macrophages (GAMs). In the GBM microenvironment, GAMs are re-programmed and lose the ability to elicit anti-tumor responses. Instead, they acquire a glioma-supporting phenotype, which is a key mechanism promoting glioma progression. Our re-analysis of published scRNA-seq data from GBM patients revealed that functional phenotypes of GAMs are linked to the level of *SORL1* expression, which was further confirmed using in vitro models. Moreover, we demonstrate that SorLA restrains secretion of TNFα from microglia to restrict the inflammatory potential of these cells. Finally, we show that loss of SorLA exacerbates the pro-inflammatory response of microglia in the murine model of glioma and suppresses tumor growth.

**Keywords** VPS10P Domain Receptors; Glioblastoma; Intracellular Sorting; Phenotypic Polarization; Brain Tumors
**Subject Categories** Cancer; Membranes & Trafficking; Signal Transduction

## Introduction

Intracellular protein sorting is essential for maintaining cellular homeostasis and activity (García-Cazorla et al, 2022). Efficient sorting of proteins in the endocytic and exocytic routes not only ensures adequate levels of receptors and transporters on the cell surface but it is also crucial for protein secretion (Yarwood et al, 2020). One of the key mechanisms governing intracellular sorting engages the VPS10P domain receptors. This receptors family entails five members (sortilin, SorCS1-3 and SorLA) which are mainly recognized for their roles in neurons (Malik and Willnow, 2020). However, in a specific pathological context, they can also be expressed in non-neuronal cells of the brain to shape their functional properties. For example, astrocytes activated after ischemic stroke express SorCS2, which is necessary for proper secretion of endostatin and for post-stroke angiogenesis (Malik et al, 2020).

SorLA, in humans encoded by *SORL1* gene, has been implicated in brain pathophysiology on genetic and functional levels. In particular, the protective role of SorLA in Alzheimer's disease (AD) has been a subject of multiple studies, focusing primarily on its neuronal functions (Malik and Willnow, 2020). However, recent data point to the potential context-dependent *SORL1* expression regulation in microglia. Thus, SNPs in *SORL1* previously associated with risk of sporadic AD possibly influence receptor gene expression in microglia rather than neurons (Nott et al, 2019). Microglial expression of *SORL1* was also noted in glioma patients' brains (Abdelfattah et al, 2022). Still, potential microglial activities of SorLA in the diseased brain are unclear.

Microglia are innate immune cells of the brain that become activated under pathological conditions. Of note, in response to particular microenvironmental cues, microglia can enter various modes of activation and acquire diverse functional properties, ranging from pro-inflammatory to immunosuppressive (Colonna and Butovsky, 2017). Although protecting the brain from insults seems to be the purpose of such activation, microglia can paradoxically also support disease progression. This is the case in glioblastoma (GBM), the common and most malignant primary brain tumor in adults (Ostrom et al, 2022). In GBM, massive accumulation of resident microglia, as well as of peripheral macrophages, is observed in the tumor microenvironment. These cells, collectively called glioma-associated microglia/macrophages (GAMs), share many functional properties, including their tumor-

[1]Faculty of Biology, University of Warsaw, 02-096 Warsaw, Poland. [2]Nencki Institute of Experimental Biology, 02-093 Warsaw, Poland. [3]Max-Delbrueck-Center for Molecular Medicine, 13125 Berlin, Germany. [4]Institute of Computer Science, 01-248 Warsaw, Poland. [5]Museum and Institute of Zoology, Polish Academy of Sciences, 00-679 Warsaw, Poland. [6]Department of (Neuro)Pathology, Academic Medical Center, University of Amsterdam, 1105AZ Amsterdam, The Netherlands. [7]Department of Biomedicine, Aarhus University, 8000 Aarhus, Denmark. [8]Stichting Epilepsie Instellingen Nederland, 2103 SW Heemstede, The Netherlands. ✉E-mail: ar.malik@uw.edu.pl

supporting phenotype. GAMs account for up to 30% of tumor mass in human GBMs and in experimental gliomas (Gieryng et al, 2017; Gabrusiewicz et al, 2016; Ochocka et al, 2021) and their role in glioma progression and immunosuppression has been shown in various glioma models (Ellert-Miklaszewska et al, 2016; Wang et al, 2012; Platten et al, 2003). Although they constitute a promising target in GBM therapy, the mechanisms shaping the functional properties of GAMs are not fully understood.

Here, we explored if SorLA plays a role in regulating the activities of GAMs by governing the intracellular sorting and secretion of its target proteins. Our in silico analysis indeed showed that *SORL1* expression in human GAMs is linked to their transcription profiles, likely reflecting diverse functional properties. Using cell models, we further demonstrate differential regulation of SorLA expression by pro-inflammatory cues and by the glioma cells. Finally, we show that SorLA acts as a sorting receptor for TNFα to limit its release from microglia. Along these lines, tumor microenvironment in SorLA-deficient mice shows enhanced pro-inflammatory properties which likely contribute to limiting glioma growth seen in these mice.

## Results

### SorLA expression levels in glioma-associated microglia/macrophages are linked to their activation mode

The activity of SorLA has been especially well documented in neurons, yet recent data suggest that it might also be expressed in other brain cell types in a specific pathological context. In particular, Abdelfattah et al reported high expression levels of the *SORL1* gene in GAMs in human samples (Abdelfattah et al, 2022). In line with this notion, we could indeed detect SorLA in Iba1+ cells in human glioma sections, although not in all studied patient samples (Fig. 1A; Appendix Table S1). Intrigued by the fact that not all Iba1+ cells in patients specimens stained positive for SorLA, we analyzed the proportions of *AIF1* (encoding for Iba1) expressing cells that also express *SORL1* in several scRNA-seq datasets from GBM samples (Data Ref.: Sankowski et al, 2019; Abdelfattah et al, 2022; Chen et al, 2021; Neftel et al, 2019; Pombo Antunes et al, 2021; Wang et al, 2022b). In all 6 datasets, *SORL1* was expressed in *AIF1*+ cells. The percentage of *SORL1*+ cells among *AIF1*+ cells was variable and ranged from 17.2% to 97.0% depending on the dataset (Appendix Table S2). Induction of *Sorl1* expression in GAMs was recapitulated in a murine model of glioma (Szulzewsky et al, 2015). Twenty days after GL261 glioma cells implantation, *Sorl1* expression in GAMs (CD11b+ cells) isolated from brain tumors increased more than three times compared to the control cells. At the same time, SorCS2, another VPS10P receptor detected in this dataset, did not show such an induction (Appendix Table S3). Taken together, these data point to the hypothesis that SorLA levels in microglia/macrophages might be upregulated during activation of these cells toward a tumor-supporting phenotype.

To further investigate *SORL1* expression patterns in GAMs, we re-analyzed scRNA-seq data from human glioma samples (Data Ref.: Abdelfattah et al, 2022), focusing on newly diagnosed GBM samples (ndGBM). From this dataset, we selected 7 clusters that we classified as GAMs based on the expression of marker genes (*AIF1, CD68, ITGAM, P2RY12, TMEM119, CX3CR1*; Fig. EV1A–C; Appendix Table S4; Dataset EV1). As reported by the authors of

the study before, *SORL1* expression was limited to this GAMs population (Fig. EV1B; Appendix Fig. S1A). The only other cluster with noticeable expression of *SORL1* was characterized by expression of interferon-inducible genes (*IFI6, IFI27, IFITM1, IFITM2, IFITM3*) as well as endothelial markers (*EDN1, ABCG2, FLT1, PLVAP, PECAM1*) (Appendix Fig. S1A; Dataset EV1).

In all subsequent steps, we focused on the population of selected glioma-associated microglia/macrophages (GAMs). These cells were pooled for further analysis to yield 23 GAMs clusters in which *SORL1* expression levels were evaluated (Fig. 1B). To shed light on the potential links between SorLA levels and the functional properties of GAMs, we next investigated the marker genes of five clusters with the highest and five clusters with the lowest *SORL1* levels (Fig. 1C; Dataset EV2). Among "high-SORL1" clusters, cluster 7 was characterized by the expression of immediate early genes, *SPP1*, a gene associated with tumor-promoting GAMs (Szulzewsky et al, 2015), and *ID2* involved in pro-tumorigenic polarization of myeloid cells (Huang et al, 2017). Cluster 22 showed high expression levels of several microglia/macrophages genes including *TREM2*, which was linked before to the pro-tumorigenic properties of tumor-associated macrophages in various cancers (Khantakova et al, 2022). At the same time, among "low-SORL1" clusters, cluster 9 was characterized by expression of *TLR* genes as well as pro-inflammatory cytokines *IL1A* and *TNF*. Clusters 11 and 17 showed relatively high expression of a pro-inflammatory factor *MIF* and several glycolysis-related genes (*PGK1, ENO1, GAPDH, LDHA, PKM*). Importantly, a metabolic switch towards glycolysis is a hallmark of pro-inflammatory activation of microglia/macrophages (Lauro and Limatola, 2020).

Furthermore, to provide a more global characteristics of GAMs, we implemented Natural Language Processing (NLP) approach, which allows to identify keywords describing functionally related groups of genes. In doing so, we analyzed marker genes of all GAMs clusters (Datasets EV2 and EV3). NLP revealed that genes enriched in "high-SORL1" clusters were associated with terms as "cancer" (clusters 2, 7, 8, 22) and "angiogenesis" (cluster 12). Concurrently, genes expressed in "low-SORL1" clusters were described with words like "glycolysis" (clusters 11 and 17), "toll-like receptor" (clusters 9 and 18) or "phagocytosis" (cluster 3). Taken together, based on the results obtained thus far we hypothesized that high and low SorLA levels might be associated with pro-tumorigenic and pro-inflammatory phenotypes of microglia/macrophages, respectively.

To gain deeper insights into the functional relevance of SorLA's presence in GAMs, we performed Monte-Carlo Feature Selection (MCFS-ID) analysis (Dramiński et al, 2008; Dramiński and Koronacki, 2018) on a single-cell level in the same dataset. MCFS-ID allows to determine features (here expression levels of a given gene) predicting the behavior of another feature (here, *SORL1* expression levels categorized to low, medium, or high). Among 25 top genes from the MCFS-ID, differential levels of gene expression in the context of discretized values of *SORL1* were assessed (Fig. 1D; Dataset EV4). This analysis highlighted microglia signature genes (*CX3CR1, A2M, C3*) and *TREM2* among the best predictors of high-SORL1 expression levels in GAMs. Moreover, high expression of a gene coding for transcription factor MEF2C, known for its role in restraining microglial inflammatory response (Deczkowska et al, 2017), was also linked to high-SORL1 transcript levels. Interestingly, we also noted a similar positive association between *SORL1* and *GPR34*, coding for a microglial receptor

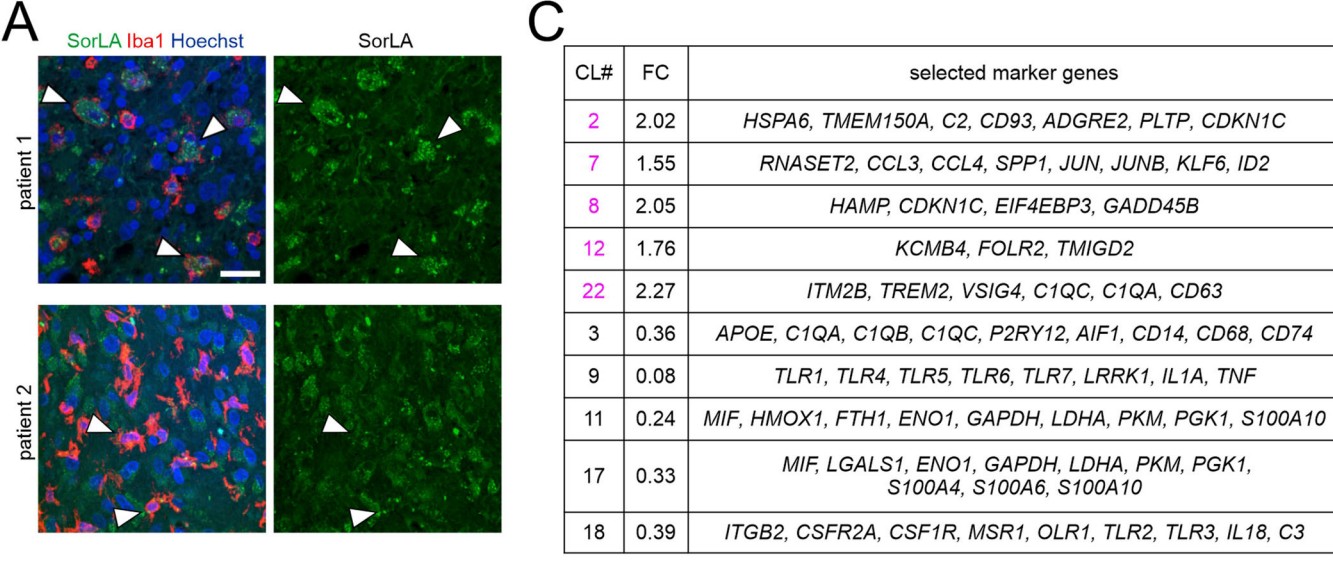

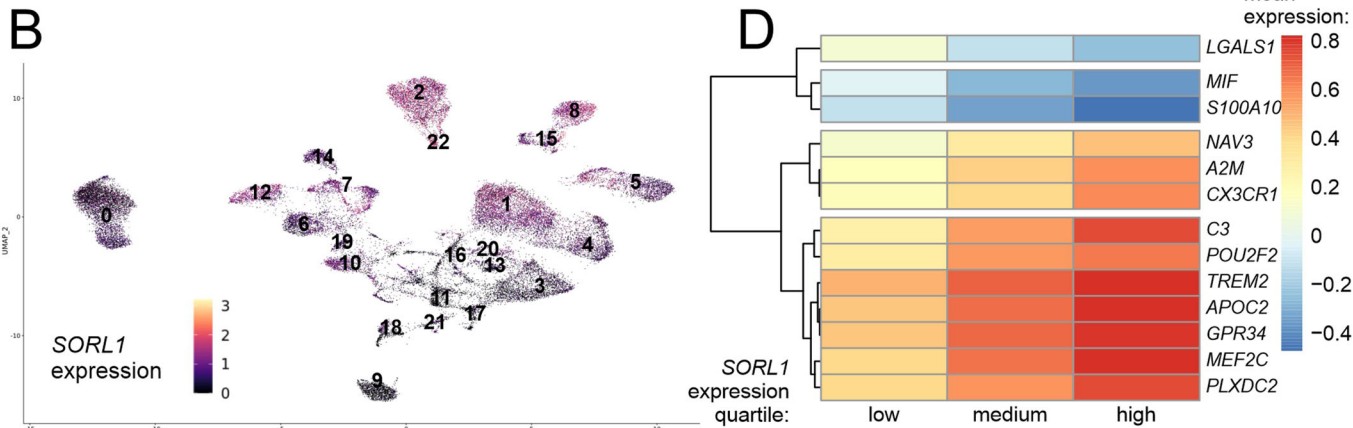

**Figure 1. SORL1 expression in human GAMs is linked to their functional properties.**

(A) SorLA is present in Iba1+ cells in some patients (#1) but absent from these cells in other cases (#2). White arrowheads indicate selected Iba1+ cells. Scale bars, 25 μm. (B) UMAP plot showing *SORL1* expression levels normalized with SCT, in clusters of human GAMs. (C) Selected marker genes of 5 GAMs clusters with the highest and 5 clusters with the lowest *SORL1* expression levels. CL#, cluster number; FC, *SORL1* expression fold change, mean cluster expression against mean expression in remaining clusters. (D) Hierarchical clustering of the most significant genes based on the mean values of standardized gene expression data. These genes were returned by MCFS-ID with the highest RI values and showed differential expression in the context of discretized values of *SORL1* gene expression.

required to maintain microglia in the homeostatic phenotype (Schöneberg et al, 2018). Finally, we identified *LGALS1, S100A10* and *MIF* as predictors of low *SORL1* expression in GAMs. While *MIF* is a well-established marker of pro-inflammatory activation of microglia/macrophages, the roles of *LGALS1* and *S100A10* in these cells are not entirely clear. *LGALS1* expression in tumor cells has been linked to immunosuppression (Chen et al, 2019), but its induction in microglia might be relevant for inflammatory responses. Thus, increased *LGALS1* expression was observed in microglia after stimulation with LPS (Kiss et al, 2023) and in a subpopulation of disease-associated microglia in multiple sclerosis (Masuda et al, 2019). S100A10 has been mostly studied in other cell types; in macrophages, it seems to be involved in their recruitment upon inflammation (O'Connell et al, 2010).

To further characterize the potential functional interactions between various cell types present in the tumor microenvironment

in the context of *SORL1* levels in GAMs, we implemented the CellChat analysis. First, we defined several cell types in the ndGBMs (Data Ref.: Abdelfattah et al, 2022) based on the expression of marker genes (Dataset EV1 and Appendix Tables S4–7). As a result, in addition to GAMs population subdivided into low-, medium- and high-*SORL1* expressing cells, we defined the populations of tumor cells (clusters 0, 7, 8, 9, 15, expressing *CDK4, MT1X, ATRX, CCND2, MDM2, SOX4, CD9, CDK6, S100B*), lymphocytes (clusters 5, 17, 19; *GZMK, CD3E, PTPRC, CCL5, IL32, CD69, CD52*), smooth muscle cells (cluster 11; *ACTA2, TAGLN*), endothelial cells (cluster 14; *EDN1 PECAM1 ANGPT2*), oligodendrocytes (cluster 12, *MBP, CNP*) and other cells whose classification was dubious (clusters 4, 18, 20). Cell–cell interaction analysis revealed distinct communication patterns specific for each of these populations (Figs. 2A,B and EV2A). Tumor cells appeared to be the major contributors to outgoing

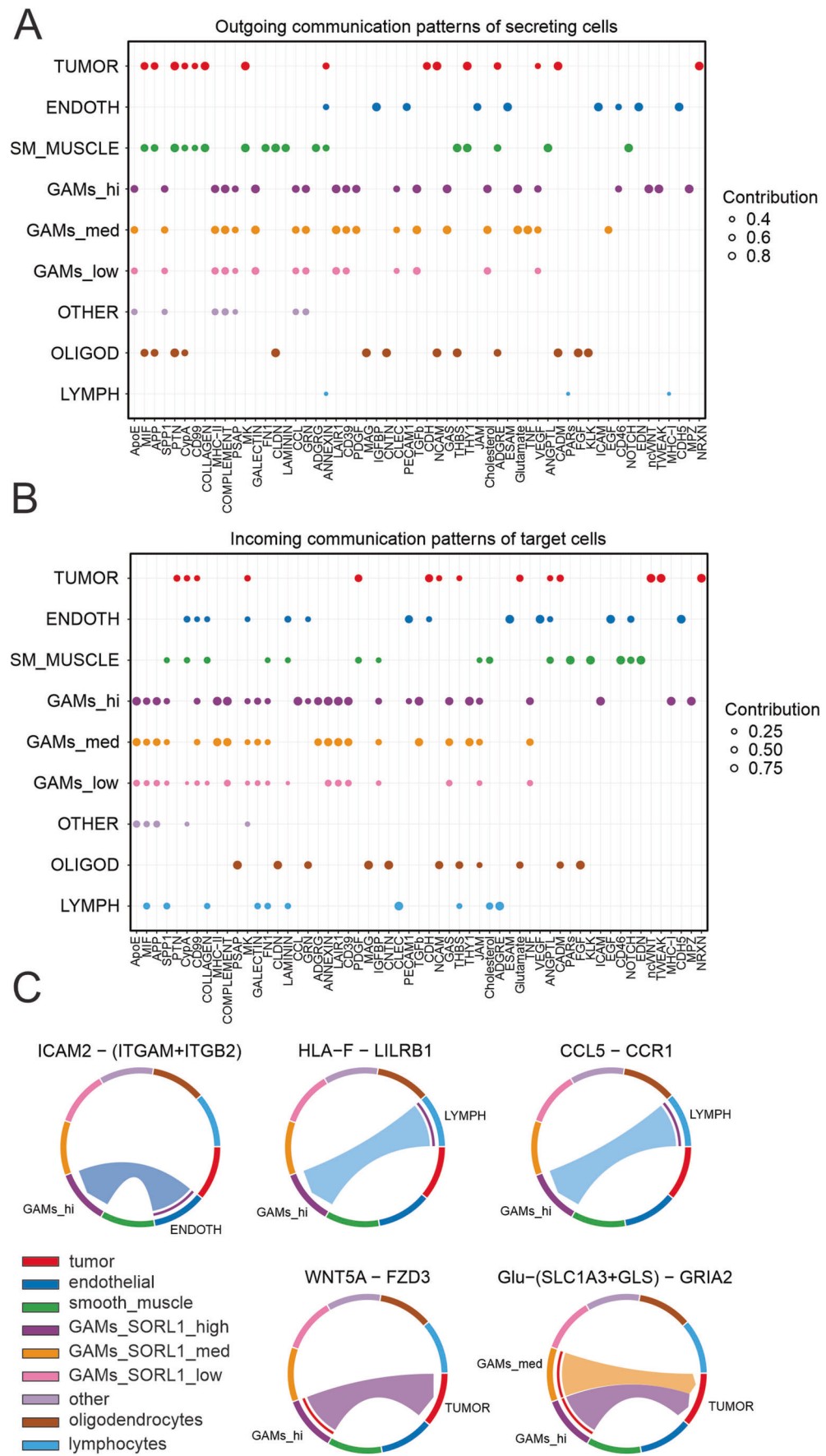

◄   **Figure 2. CellChat analysis reveals distinct interaction networks between cell populations in newly diagnosed GBM tumors.**

(A, B) Dot plots showing relative contributions of signaling pathways to the outgoing signaling patterns of secreting cells (A) and incoming signaling patterns of target cells (B) for distinct cell populations inferred from ndGBM in Abdelfattah et al. The dot size is proportional to the strength of the contribution score. The higher contribution score, the higher enrichment of the signaling pathways in the corresponding cell group. Tumor cells (tumor), endothelial cells (endoth), smooth muscle cells (sm_muscle), GAMs, oligodendrocytes (oligod) and lymphocytes (lymph) were selected based on expression of marker genes (Appendix Tables S4–S7; Dataset EV1). GAMs with low, medium and high-*SORL1* expression levels (GAMs_low, GAMs_med, GAMs_hi) were separated prior to analysis. (C) Chord diagrams indicating selected ligand-receptor pairs mediating interaction between cell populations. The width of chords is proportional to signal strength of a given ligand-receptor pair.

communication, which at the same time was almost missing for the lymphocytes that mostly exhibited incoming patterns. Of note, GAMs tended to show different communication patterns depending on their *SORL1* expression levels (Figs. 2A,B and EV2A).

More detailed analysis of the ligand-receptor pairs contributing to these communication patterns revealed several interactions specific for the GAMs with high-*SORL1* expression. This population received CCL5-CCR1 signals from the lymphocytes and ICAM2-(ITGAM + ITGB2) from the endothelial cells, as well as HLA-F–LILRB1 from the lymphocytes (Fig. 2C). These interactions may be associated with enhanced tissue infiltration (Schenkel et al, 2004; Pham et al, 2012) and induction of immunosuppressive GAMs phenotype (Zeller et al, 2023), respectively. In turn, high-*SORL1* GAMs sent pro-tumorigenic signals to the tumor cells (Fig. 2C). In particular, noncanonical Wnt signaling (WNT5A-FZD3) might promote glioma invasion (Pukrop et al, 2010, 2006) and glutamatergic signaling ((SLC1A3 + GLS)–GRIA2) is known to stimulate tumor cells growth, proliferation, and survival (Prickett and Samuels, 2012). Other ligand-receptor pairs revealed by this analysis included TNF-TNFRSF1A sent by medium-*SORL1* expressing GAMs, TNFSF12-TNFRSF12A (high-*SORL1* GAMs to tumor cells), PDGFB-PDGFRA (medium- and high-*SORL1* GAMs to several populations), and THY1-(ITGAM + ITGB2) received by high-*SORL1* GAMs (Fig. EV2B). In conclusion, CellChat analysis further substantiated our notion that high expression of *SORL1* is one of the features characterizing a distinct GAMs population whose transcriptional profiles point to their pro-tumorigenic potential.

Finally, we repeated key analyzes on an independent dataset published by Neftel et al, (Data Ref.: Neftel et al, 2019). Here, *SORL1* expression was also seen predominantly in GAMs that we selected based on the expression of the marker genes as before for the Abdelfattah et al data (Dataset EV5; Fig. EV1D,F; Appendix Fig. S1B; Appendix Table S8), but it appeared more uniform and shifted towards higher levels compared to the dataset from Abdelfattah et al (Appendix Fig. S1C,D). In the next step, GAMs were further grouped into 5 clusters, for which *SORL1* expression levels and the marker genes were analyzed (Appendix Fig. S1E–H; Dataset EV6). Marker genes of the two GAMs clusters with the highest *SORL1* expression included *CX3CR1*, *P2RY12*, *AIF1*, *ITGAM*, *TMEM119*, *TREM2*, *CCL3*, and *CCL4*, while GAMs cluster with the lowest *SORL1* levels was characterized by *LGALS1*, *GAPDH*, *PGK1*, *ENO1*, *LDHA*, *MIF*, *FTH1*, *HMOX1*, and *TLR4* marker genes (Appendix Fig. S1H). Moreover, the genes that we previously linked to *SORL1* expression on a single-cell level (Abdelfattah et al data, Fig. 1D) showed similar expression patterns associated with *SORL1* in this additional scRNA-seq dataset (Appendix Fig. S1I). In conclusion, our key findings on *SORL1* expression patterns in GAMs were recapitulated in this independent analysis.

Taken together, we propose a functional link between the activation status of microglia/macrophages and *SORL1* expression levels. In particular, our results point to the scenario where high *SORL1* expression occurs in tumor-supportive GAMs, while low *SORL1* expression is associated with pro-inflammatory phenotypes of microglia/macrophages.

## Loss of SorLA promotes TNFα release from cultured microglia

Since *SORL1* expression appeared related to the functional properties of GAMs, we tested whether SorLA levels might be specifically regulated by the cues triggering diverse microglial phenotypes. To study this phenomenon, we used primary murine microglia treated with LPS or co-cultured with GL261 glioma cells, an in vitro model to mimic pro-inflammatory stimulation and the impact of glioma-secreted factors, respectively. In line with our hypothesis, *Sorl1* expression increased in the presence of glioma cells, while it dramatically decreased upon LPS treatment (Fig. 3A; Appendix Fig. S2A,B).

Distinct changes in microglial *Sorl1* expression seen upon activation by LPS and in the presence of glioma cells suggested that SorLA might be an active player in shaping functional properties of microglia. SorLA controls the intracellular sorting of target proteins defining plasma membrane transport and secretion properties (Schmidt et al, 2017). As cytokines release from activated microglia is crucial for their activity and response to disease (Colonna and Butovsky, 2017), we profiled cytokines released by WT and SorLA-deficient (SorLA-KO, SLKO) murine microglia upon PMA stimulation to uncover potential factors secreted in a SorLA-dependent manner. We did not observe any global alterations in cytokines secretion from SorLA-KO microglia as compared to WT cells. Several cytokines were released in similar amounts in both genotypes, including IL-9, MCP1, MIP1α, MIP2, and MIP3 (Fig. EV3). Secretion of MIP1β tended to be decreased in SorLA-KO cells. However, the most remarkable difference was seen in the secretion of the pro-inflammatory cytokine TNFα, which was released in higher amounts from SorLA-deficient microglia (Fig. 3B). This enhanced secretion of TNFα was not due to its increased expression, as mRNA levels were not changed in SorLA-KO cells (Fig. 3C; Appendix Fig. S2C). These results indicated that the alterations in TNFα release occur post-transcriptionally and might result directly from SorLA-dependent sorting mechanisms present in WTs, but absent in SorLA-KO microglia.

To further establish the relevance of our findings for human cells, we used microglia derived from induced pluripotent stem (iPS) cells (Fig. 3D,E) either wild-type or genetically deficient for *SORL1* (SLKO). As expected, expression of pluripotency markers (*SOX2*, *NANOG*, *OCT4*) dropped during differentiation, while microglia markers (*P2RY12*, *TREM2*, *AIF1*, *CX3CR1*, *ITGAM*)

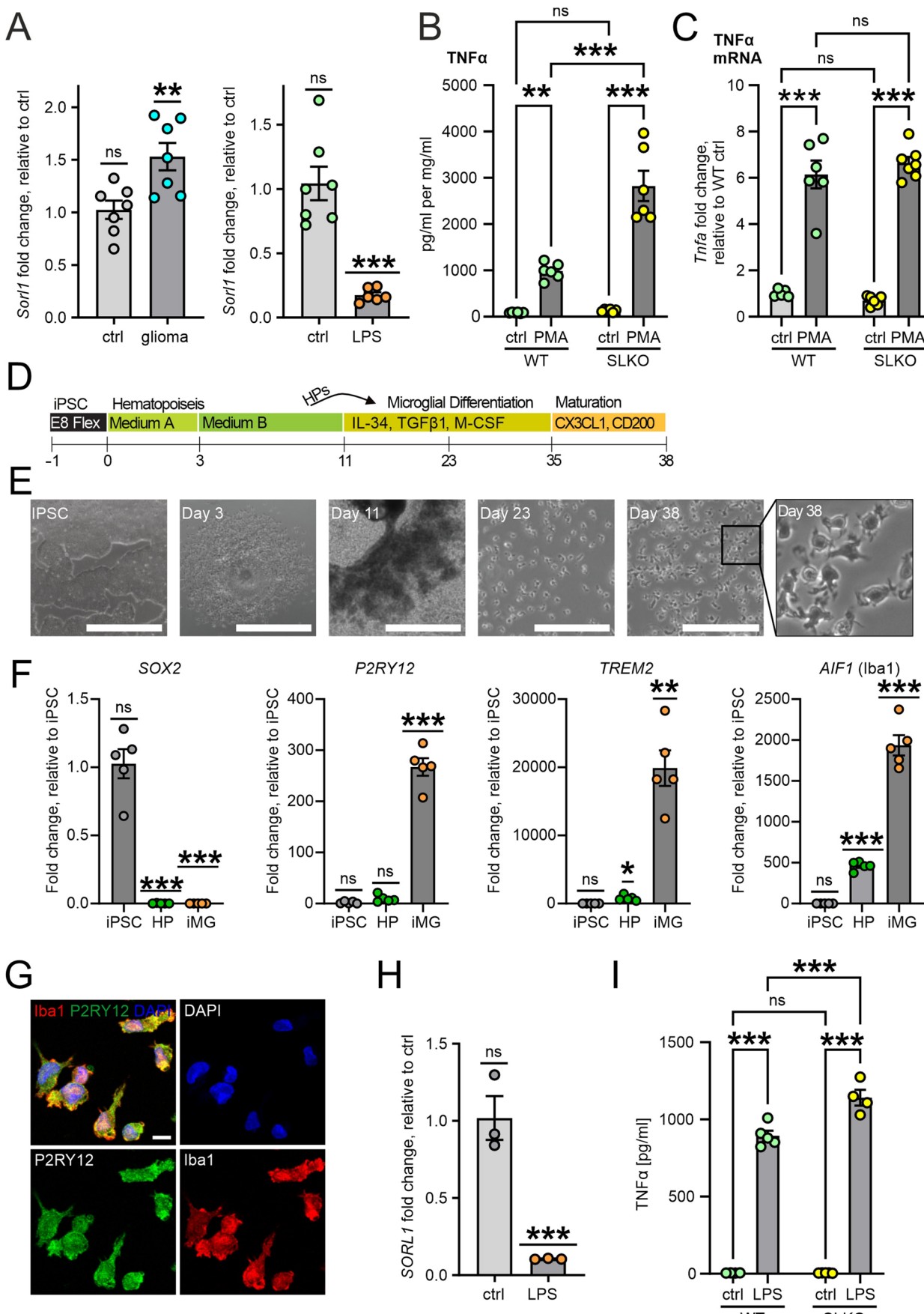

**Figure 3.  SorLA is differentially regulated by pro- and anti-inflammatory cues and restricts TNFα release from microglia.**

(A) *Sorl1* mRNA levels in primary murine microglia co-cultured with glioma cells (left) or stimulated with LPS (right) as assessed by qRT-PCR (relative to *Hprt1* or *β2M*, respectively). $n = 6$–7 biological replicates. (B) TNFα levels as determined by ELISA in cell culture medium from primary WT and SorLA-KO microglia either untreated (ctrl) or treated with PMA for 24 h. TNFα levels were normalized to the protein content in the respective cell lysates. $n = 6$ biological replicates. (C) *Tnfa* mRNA levels in primary murine microglia stimulated with PMA as assessed by qRT-PCR (relative to *Hprt1*). $n = 5$–7 biological replicates. (D) Outline of the iPSC-to-microglia (iMG) differentiation protocol. (E) Phase contrast images of iPSCs, HPs, and iMG at different stages of the microglia differentiation. Scale bar, 1000 μm (iPSC, day 3, day 11); 200 μm (day 23, day 38). (F) Expression levels of marker genes for pluripotent stem cells (*SOX2*) and microglia (*P2RY12, TREM2, AIF1*) in iPSCs, HPs and iMG during microglia differentiation as assessed by qRT-PCR (relative to *GAPDH*). $n = 5$ biological replicates. (G) Representative images of human iMG immunostained for microglia markers Iba1 (red) and P2RY12 (green) and counterstained with DAPI (blue). Scale bar, 10 μm. (H) *SORL1* mRNA levels in human iMG stimulated with LPS as assessed by qRT-PCR (relative to *β2M*). $n = 3$ biological replicates. (I) TNFα levels as determined by ELISA in cell culture medium from WT and SorLA-KO iMG either untreated (ctrl) or treated with LPS for 24 h. $n = 4$–5 biological replicates. Data information: (A–C, F, H, I) Data are presented as mean ± SEM. ns not significant; *$P < 0.05$; **$P < 0.01$; ***$P < 0.001$ in one-sample *t* test compared to 1 (A, F, H) or in two-way ANOVA with Tukey's multiple comparisons (B, C, I). Source data are available online for this figure.

levels increased (Fig. 3F; Appendix Figs. S2D and S3A). Immunodetection of Iba1 and P2RY12 additionally confirmed the microglial identity of generated cells (Fig. 3G). Differentiation was unaltered by SorLA deficiency, as expression levels of marker genes *AIF1*, *ITGAM* and *P2RY12* were comparable for WT and SorLA-deficient human-induced microglia (iMG, Appendix Figs. S2E and S3B). As anticipated, SorLA protein was completely lost from SLKO iMG (Appendix Fig. S3C). Using this model, we confirmed that LPS stimulation drives a remarkable decrease in *SORL1* expression in WT iMG (Fig. 3H; Appendix Fig. 2F). Moreover, loss of SorLA activity led to an increased TNFα release from iMG (Fig. 3I), indicating that SorLA-dependent control of TNFα secretion is conserved between the species and highlighting its potential relevance for disease pathogenesis.

## SorLA binds TNFα to control its intracellular trafficking

Typically, SorLA exerts its functions by binding target proteins and directing their intracellular trafficking. For example, this sorting activity of SorLA was documented for protein sorting between the TGN, endosomes and lysosomes, as well as in the recycling route via the Rab11+ compartment (Schmidt et al, 2016, 2007; Caglayan et al, 2014). To corroborate the potential role of SorLA in TNFα sorting, we first tested the colocalization of the two proteins in microglial cells. Indeed, immunostaining of PMA-stimulated BV2 cells revealed partial overlap of SorLA and TNFα signals (Fig. 4A,B). Next, we examined the interaction of SorLA with TNFα in co-immunoprecipitation (co-IP) assays. Towards this end, we over-expressed SorLA and GFP-tagged TNFα (or GFP alone) in HEK293 cells and pulled down the GFP tag. SorLA was present in the immunoprecipitate containing TNFα-GFP, while it was not visible in the control GFP-IP (Fig. 4C), supporting our notion that TNFα is a SorLA ligand.

As the structure of SorLA entails several domains capable of cargo binding (Fig. 4D), we sought to identify the domain responsible for the interaction with TNFα. Using deletion mutants lacking particular SorLA domains (Fig. EV4A) in our co-IP experiments, we observed that removing EGF-type repeat and the β-propeller (ΔEGF/βP mutant) tended to weaken SorLA binding to TNFα, although these results did not reach statistical significance (Fig. EV4B,C). Of note, these experiments did not rule out additional binding sites outside the EGF-type repeat and the β-propeller for TNFα in SorLA. Using more stringent co-IP conditions (300 mM NaCl) in order to increase the specificity of

our results led to loss of binding of TNFα by full-length SorLA (Appendix Fig. S4). As an alternative approach, we used myc-tagged mini-receptors composed exclusively of particular SorLA domains (Fig. EV4A; Appendix Table S9). In line with our prior observations, the most efficient co-IP with TNFα was noted for the mini-receptor encompassing the EGF-type repeat and β-propeller (EGF/βP, Fig. 4E,F). In summary, it is plausible that SorLA binds TNFα predominantly via an extracellular motif containing EGF-type repeat and β-propeller.

Finally, we elucidated the impact of SorLA on the intracellular trafficking of TNFα. We analyzed its colocalization with the markers of subcellular compartments in WT and SorLA-deficient primary microglia stimulated with PMA. We did not observe remarkable TNFα presence in lysosomes (stained with anti-Lamp1) in any of the genotypes (Appendix Fig. S5). Rather, TNFα was colocalizing with the Golgi (GM130-positive structures), Rab7+ late-endosomes, Rab11+ recycling endosomes, as well as with Vti1b, known for its important role in the TNFα secretory route (Murray et al, 2005). SorLA deficiency did not affect the presence of TNFα in the GM130+, Lamp1+ and Rab7+ compartments (Appendix Fig. S5), but it increased the colocalization of TNFα with Vti1b and caused a concurrent loss of TNFα from Rab11+ endosomes (Fig. 4G). These results indicated that loss of SorLA shifts the trafficking of TNFα toward the secretory pathway, which could explain the increased TNFα release from SorLA-KO microglia.

## Loss of SorLA limits glioma growth, promotes inflammation and necroptosis

SorLA emerged as a critical factor controlling the functional properties of microglia, restricting their pro-inflammatory activities. Moreover, in glioma patients, *SORL1* expression levels in GAMs were related to their transcription profiles. We further speculated that the presence of SorLA might have an impact on the functional properties of GAMs and, consequently, on tumor microenvironment and glioma progression.

Thus, we asked whether SorLA presence in the host cells has an impact on glioma progression in a murine model. Glioma GL261 cells carrying luciferase and tdTomato transgenes were implanted to the striata of WT and SorLA-KO mice, and the tumor growth was followed for 21 days. Using this model, we showed that glioma growth is limited in SorLA-deficient mice (Fig. 5A,B), supporting our notion that the presence of this sorting receptor in

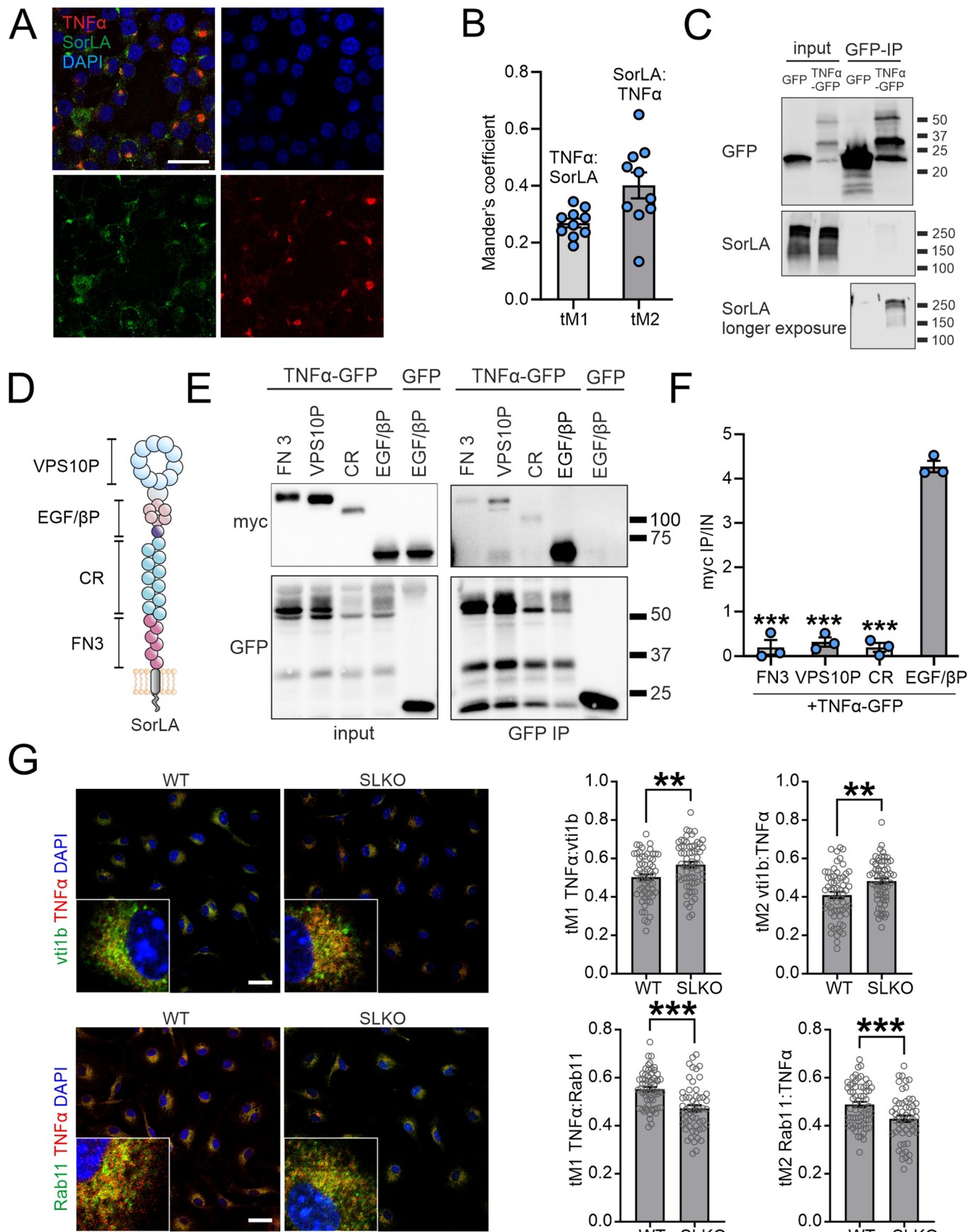

**Figure 4. SorLA interacts with TNFα to regulate its subcellular distribution.**

(A) Representative image of SorLA and TNFα immunostaining in BV2 microglial cells stimulated with PMA. Cells were counterstained with DAPI. Scale bar: 25 µm. (B) Results of colocalization analysis performed for SorLA and TNFα signals exemplified in (A) calculated as thresholded Manders coefficients (tM). n = 10 fields of view. (C) Co-immunoprecipitation (co-IP) of SorLA with GFP-tagged TNFα overexpressed in HEK293 cells after GFP-IP. GFP serves as a negative control. (D) Scheme of SorLA protein structure indicating its functional domains. VPS10P VPS10P domain, EGF/βP EGF-type repeat/β-propeller domain, CR complement-type repeat, FN3 fibronectin-type III domain. (E) TNFα preferentially binds the EGF/β-propeller SorLA mini-receptor. Co-immunoprecipitation (co-IP) of myc-tagged SorLA mini-receptors with TNFα-GFP overexpressed in HEK293 cells after GFP-IP. GFP serves as a negative control. (F) Ratios of myc signals in co-IP and input samples (IP/IN) calculated for each transfection variant as in (E). n = 3 biological replicates. (G) Immunostaining for TNFα and the markers of secretory vesicles (Vti1b) and recycling endosome (Rab11) in WT and SorLA-KO microglia. Cells were counterstained with DAPI. Representative images (left) and the results of colocalization analysis (right) calculated as thresholded Manders coefficients (tM) are shown. n = 55–64 cells. Scale bar, 25 µm. Data information: (B, F, G) Data are shown as mean ± SEM. **P < 0.01; ***P < 0.001 in one-way ANOVA with Tukey's multiple comparisons test (compared to EGF/βP; F) or in unpaired two-tailed t test (G). No statistical analysis was performed for (B) as we did not intend to directly compare tM1 and tM2. Source data are available online for this figure.

host cells is critical for establishing a tumor-promoting microenvironment.

To evaluate the combined impact of the developing glioma and SorLA loss from host cells on the tumor microenvironment, we first assayed the levels of selected cytokines (TNFα, MIP2, CCL5, CXCL1, IL1β, IL2, IL6, and IL10) in the tissue lysates derived from tumor-bearing and tumor-free hemispheres. At 14 days post-implantation, we did not document any major induction of the cytokines (Appendix Fig. S6A). Twenty-one days after implantation, induction of all cytokines in glioma-bearing hemispheres was visible and comparable between genotypes, except for MIP2, which was decreased in the SLKO tumor samples as compared to WTs (Appendix Fig. S6B).

Next, we evaluated the properties of microglia in the glioma model in WT and SLKO mice. The microglia activation status is reflected by the changes in their morphology (Morrison et al, 2017; Franco-Bocanegra et al, 2021; Kvisten et al, 2019). In essence, the tumor-supportive phenotype is characterized by ramified morphology with longer and more branched processes. By contrast, pro-inflammatory microglia present compact morphology and shorter extensions. These features can be evaluated by Sholl analysis, which quantifies the number of processes crossing the spheres of increasing radius, centered at the cell soma. This analysis performed on the tumor-surrounding Tmem119+ cells revealed remarkable differences between WT and SorLA-KO microglia (Fig. 5C,D). In detail, SorLA-KO microglia showed a compact morphology with shorter processes, which can be attributed to its more pro-inflammatory phenotype, while a branched morphology of microglia observed in WTs can correspond to a homeostatic or tumor-supporting state. These differences between the genotypes were also reflected by an apparent global decrease in Tmem119 signal in the tumor-surrounding tissue in SorLA-KO mice (Appendix Fig. S7A,B). Of note, in the contralateral glioma-free hemisphere, we did not observe any genotype-dependent alterations in microglia morphology (Appendix Fig. S7C,D).

To further verify the hypothesis that the pro-tumorigenic activities are blunted in the SorLA-KO animals, we focused on the phosphorylation status of STAT3 in the glioma-bearing brains. STAT3 is a transcription factor, which, when phosphorylated, activates the expression of multiple genes related to pro-tumorigenic properties of GAMs (De Boeck et al, 2020; Dumas et al, 2020). It was demonstrated that inhibition of STAT3 enhances pro-inflammatory potential of these cells, which results in suppression of tumor growth in a murine model of glioma (Zhang et al, 2009). In line with our hypothesis, the levels of p-STAT3 were remarkably reduced in glioma-bearing hemispheres of SorLA-deficient mice as compared to WTs (Fig. 5E,F).

An important consequence of the inflammatory response is the influx of peripheral immune cells into the affected tissue. Locally released factors attract circulating leukocytes and promote their migration, eventually driving their infiltration into the inflamed area. In GL261 gliomas, the infiltration of galectin-3+ macrophages and CD8 + T lymphocytes into the tumor mass was similar for both WT and SorLA-KO mice (Fig. 6A–C). Also the staining for the common GAMs marker Iba1 did not reveal any genotype-dependent differences in terms of glioma infiltration (Appendix Fig. S8). However, we noted a striking genotype-dependent difference in the neutrophil influx into the glioma. Thus, infiltration of the MPO+ neutrophils was evident in the gliomas in SorLA-KO brains, while it did not occur in the WTs and in the contralateral hemispheres (Fig. 6D,E). This was not due to the overall increase in neutrophil amounts in the circulating blood of SorLA-KO mice, as the numbers of circulating neutrophils, as well as of erythrocytes, monocytes, lymphocytes, basophils, and eosinophils, were comparable in glioma-bearing WT and SorLA-KO mice (Appendix Fig. S9). We propose that this massive neutrophil infiltration is a direct consequence of a pro-inflammatory milieu promoting their migration into the brain parenchyma in SorLA-deficient mice. One of the key mechanisms facilitating neutrophils influx into the tissue involves TNFα-driven induction of ICAM-1, an adhesion molecule critical for the transendothelial migration of these cells (Peterson et al, 2006; Yang et al, 2005). Increased soluble ICAM-1 (sICAM1) levels are also a well-established hallmark of inflammation (Bui et al, 2020). As anticipated from our data, sICAM1 was elevated in the tumor-containing hemispheres derived from SorLA-KO mice as compared to the WTs (Fig. 6F). These results strongly supported our hypothesis that loss of SorLA shifts the properties of the glioma microenvironment toward pro-inflammatory.

Finally, to further elucidate the mechanisms limiting tumor growth in SorLA-KO mice, we focused on cell death mechanisms that might be activated by TNFα itself, or by the infiltrating neutrophils. TNFα can trigger apoptosis or necroptosis via its receptor TNFR1 (Webster and Vucic, 2020), while neutrophils elicit ferroptosis (Yee et al, 2020). We thus checked which of these pathway(s) are activated in glioma specifically in SorLA-KO mice. We did not observe induction of apoptosis, as the cleavage of PARP and caspase-3 was negligible and similar for both WT and SorLA-deficient mice (Fig. 7A,B). Induction of ferroptosis was not visible either (Fig. 7C). At the same time, we noted increased levels of

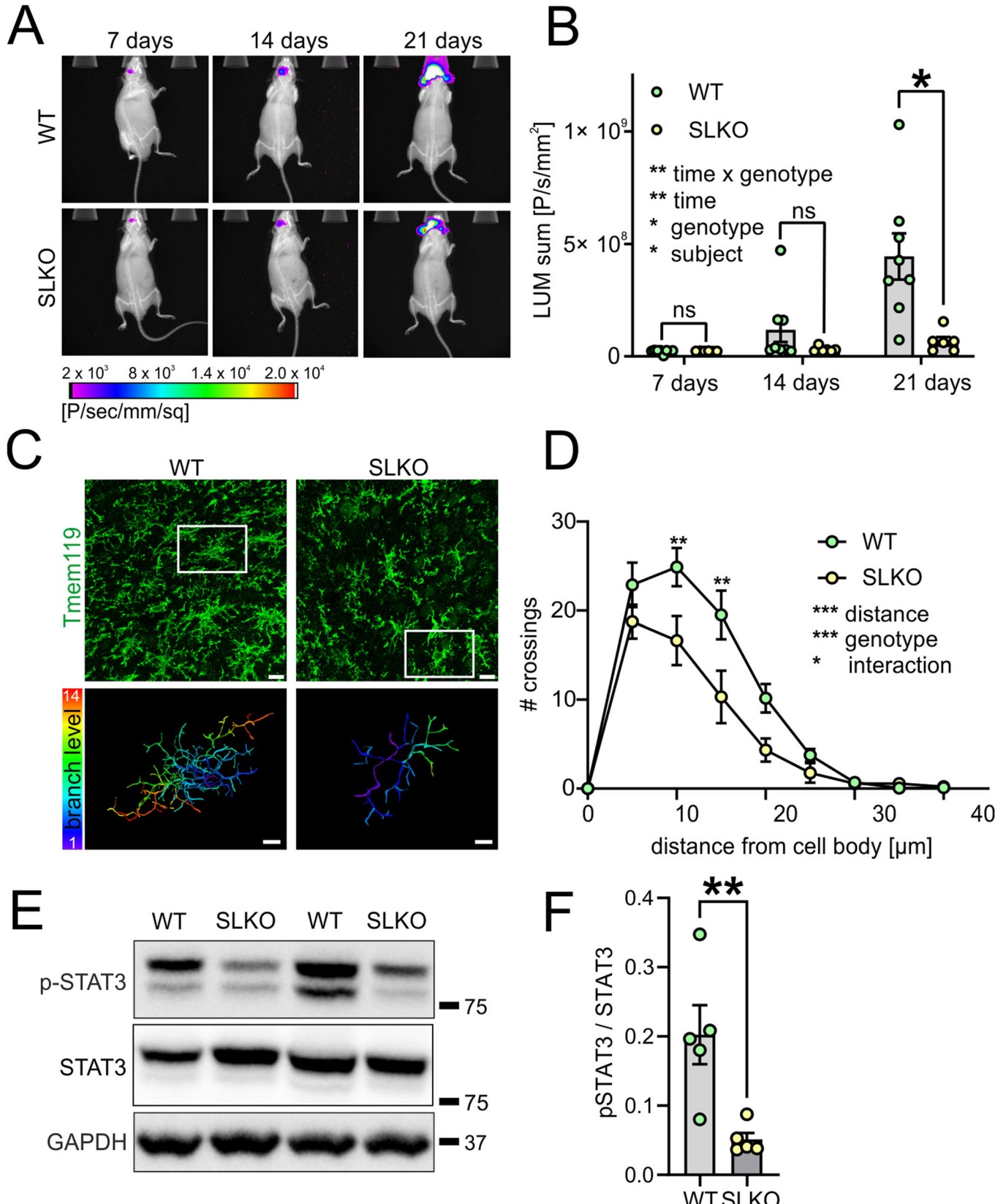

**Figure 5. SorLA deficiency inhibits glioma growth and promotes pro-inflammatory properties of microglia.**

(A) Representative images of bioluminescence signals emitted by luciferase-expressing gliomas in WT and SorLA-KO mice at 7, 14, and 21 days post-implantation. Relative signal intensities represented by color are combined with X-ray images. (B) Bioluminescence signals measured as in (A) at indicated days post-implantation. $n = 6$–8 mice per genotype. (C) Upper panel: representative images of microglia morphology revealed by Tmem119 staining in WT and SorLA-KO mice in glioma margin 21 days post-implantation. Scale bar, 15 µm. The white box indicates the cell reconstructed below. Lower panel shows reconstructed microglia branches; color depicts branch level. Scale bar, 5 µm. (D) Sholl analysis of microglia morphology reconstructed as in (C). $n = 4$ mice per genotype; for each mouse, five cells were quantified and an average of obtained values was treated as an individual data point. (E) Western blot analysis of p-STAT3 levels in WT and SorLA-KO glioma-bearing hemispheres at 21 days post-implantation. STAT3 and GAPDH are detected as loading controls. (F) Quantification of the western blot analysis as in (E). Signal intensities for p-STAT3 were normalized to STAT3 signals. $n = 5$ mice per genotype. Data information: (B, D, F) Data are shown as mean ± SEM. ns not significant; *$P < 0.05$; **$P < 0.01$; ***$P < 0.001$ in repeated measures two-way ANOVA with Sidak's multiple comparisons test (B), in two-way ANOVA with Sidak's multiple comparison test (D) or in unpaired two-tailed $t$ test (F). Source data are available online for this figure.

necroptosis markers p-RIP1 and p-RIP3 in the glioma-bearing hemispheres from SorLA-KO mice, as compared to the WTs (Fig. 6D,E). In line with these observations, we further documented a time-dependent increase of p-RIP1 in cultured glioma cells treated with TNFα (Fig. 7F). These results suggested that necroptosis might contribute to the elimination of glioma cells and, in consequence, to limiting tumor growth in SorLA-KO brains.

## Discussion

SorLA is an important player in maintaining the functional integrity of the brain (Malik and Willnow, 2020). Although this role has mainly been attributed to its neuronal functions in the past, single-cell sequencing approaches discovered the complexity of SorLA expression patterns in the diseased brain. In particular, potential microglial activities of SorLA are gaining increasing attention due to relatively high expression levels of the receptor in this cell type (Olah et al, 2018; Sankowski et al, 2019) and to the relevance of microglial activity to the pathogenesis of virtually all brain disorders (Prinz et al, 2019). We have focused on the microglial roles of SorLA in the context of glioma. However, it is plausible that such a function also bears relevance for other brain diseases, such as AD, and that the ability of SorLA to limit pro-inflammatory activity of microglia shown in this study may represent a mechanism of fundamental significance.

Here we show that microglial SorLA levels are differentially modulated in the context of inflammation and in glioma-induced immunosuppression. This regulatory mechanism has important implications for the functional properties of these cells, as SorLA acts as a sorting receptor for TNFα. We propose that SorLA-mediated restriction of TNFα release results in blunting pro-inflammatory activities of SorLA+ cells and modulates the properties of the brain microenvironment (Fig. 8).

In line with this notion, we discovered that the response of SorLA-KO mice to glioma growth is shifted towards a pro-inflammatory state as compared to WT animals. This phenotype is manifested by morphological changes of microglial cells, reduced p-STAT3 and increased sICAM1 levels, and by neutrophils infiltration. Importantly, the lack of SorLA did not influence the numbers of circulating neutrophils in glioma-bearing mice, nor did it drive neutrophils infiltration to the tumor-free brain hemisphere. This observation strongly favors the hypothesis that neutrophils' infiltration is a secondary consequence of the pro-inflammatory character of the tumor microenvironment in SorLA-KO mice.

We further propose that GAMs polarization towards the anti-tumor state in SorLA-KO brains is the plausible cause for the reduced tumor growth. In our view, TNFα release from SorLA-deficient microglia promotes necroptosis of glioma cells and enhances neutrophil infiltration. We assume that these factors are associated with the overall anti-tumor response that leads to beneficial effects in the glioma microenvironment of genetically modified animals. Although the role of inflammation in tumor biology remains a matter of debate (Montfort et al, 2019; Josephs et al, 2018), depleting TNFα in host cells resulted in larger tumors and shorter survival in the murine glioma model (Villeneuve et al, 2005). In line with this notion, inhibition of STAT3 in a murine model of glioma enhanced TNFα expression in microglia/macrophages, blocked tumor growth and improved survival (Zhang et al, 2009). Also in GBM patients, high levels of the cytokine, both on the periphery and in the tumor microenvironment were linked to increased overall survival (Cavalheiro et al, 2023).

Likewise, the role of neutrophils in tumor progression remains unclear. In several studies their inflammatory activation has been linked to poor GBM patient outcomes (Wang et al, 2022a), while experiments on murine models of glioma indicated that neutrophils can limit tumor growth in the early stage of the disease (Magod et al, 2021). A similar phenomenon was observed in metastasis, where tumor-entrained neutrophils inhibited spreading of cancer cells into other tissues (Granot et al, 2011; López-Lago et al, 2013). Thus, neutrophils polarization into pro- or anti-tumor phenotypes seems to be highly context-dependent (Lin et al, 2021; Friedmann-Morvinski and Hambardzumyan, 2023). Furthermore, direct role of TNFα in shaping neutrophils properties remains elusive, as it has been shown that this cytokine may induce their pro-tumorigenic (Maas et al, 2023) or anti-tumoral (Finisguerra et al, 2015) potential. In summary, we speculate that the pro-inflammatory glioma microenvironment of SorLA-KO mice not only drives neutrophils recruitment, but also promotes their anti-tumorigenic functions. However, it needs to be noted that we observed neutrophils influx into the SorLA-KO glioma microenvironment in the late stage of the disease and we did not investigate the properties of these cells in detail. Therefore, further studies are required to conclude about the role of neutrophils in gliomas of SorLA-depleted mice.

Overall, unlocking the endogenous pro-inflammatory potential of microglia/macrophages directly at the site of glioma may represent a potent mechanism of limiting its growth. In this context, targeting SORL1 expression in GAMs or the interaction between SorLA and TNFα emerges as an exciting strategy for

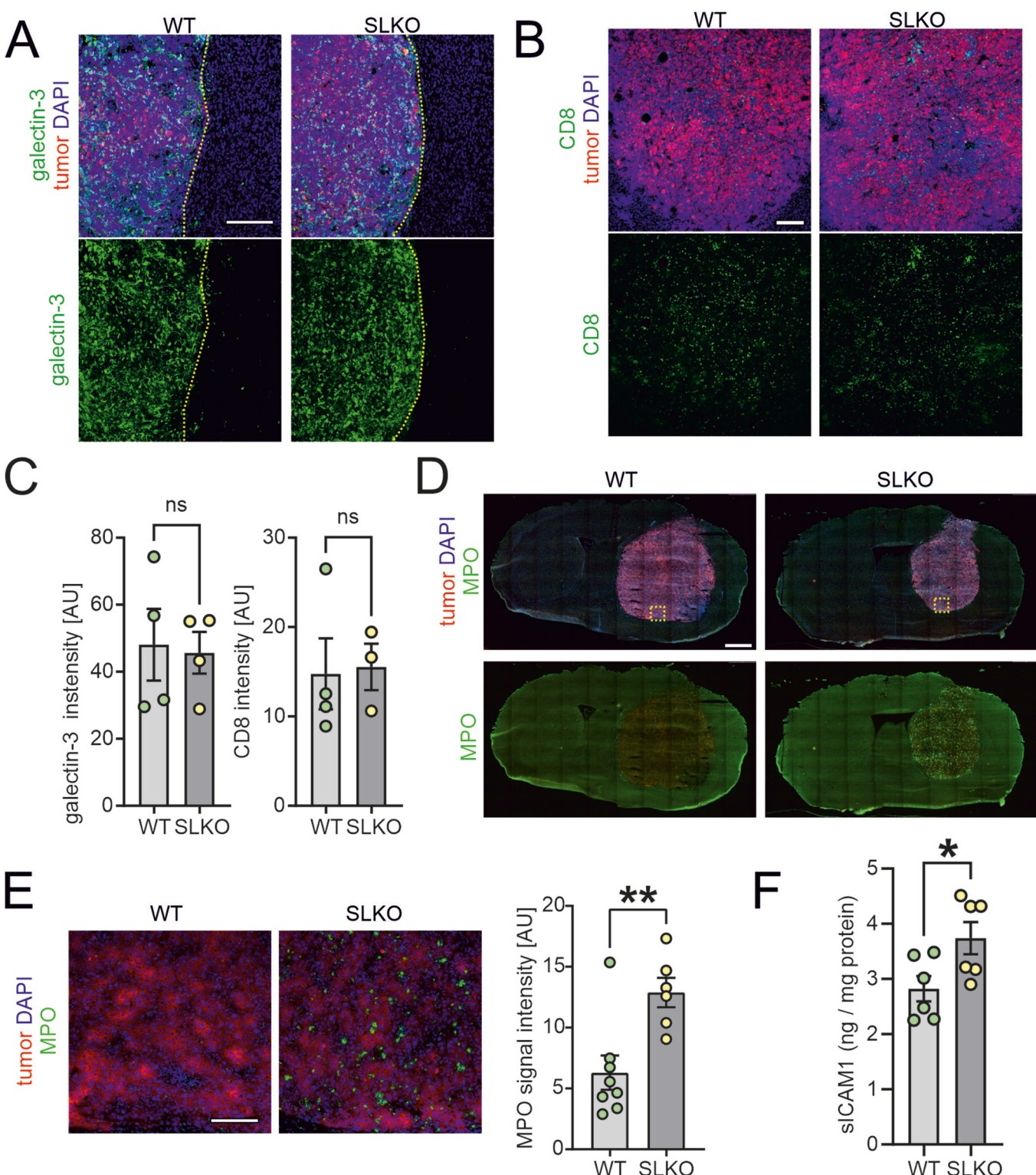

future pharmacological interventions in glioma. However, it remains unclear whether reducing SorLA levels in GAMs, although resulting in reduced size of tumors, would be ultimately beneficial for the patients. At present, no data exist that would allow for analysis of the potential association between *SORL1* expression in GAMs and GBM patient's survival. Nevertheless, the expression of *SORL1* in human GAMs is evident. Further studies on the molecular details of the mechanism described herein and on its pathophysiological implications are needed to evaluate its clinical relevance.

◄ **Figure 6.  Glioma microenvironment of SorLA-KO mice is infiltrated by neutrophils.**

(A, B) Representative images of the sections from glioma-bearing WT and SorLA-KO brains 21 days after implantation of GL261-tdTomato+Luc+ cells, immunostained for the markers of macrophages, galectin-3 (A) and cytotoxic T lymphocytes, CD8 (B). Tumor cells are seen in red. Yellow dotted line marks tumor border. Sections were counterstained with DAPI (blue). Scale bars, 200 μm. (C) Galectin-3 and CD8 signal intensities in WT and SorLA-KO glioma-bearing hemispheres. $n = 4$ mice per genotype. (D) Representative images of murine brains sections 21 days post-implantation stained for neutrophils marker MPO and counterstained with DAPI (blue). Tumor cells are seen in red. Scale bar, 1 mm. Yellow box indicates the area imaged with higher magnification in (E). (E) MPO+ cells in the glioma mass in WT and SorLA-KO brains. Scale bar, 100 μm. Right panel: quantification of MPO signal intensity observed in WT and SLKO mice. $n = 6$–8 mice per genotype. (F) Levels of sICAM1 in soluble fractions extracted from glioma-bearing brain hemispheres 21 days post-implantation, normalized to protein content. $n = 6$ mice per genotype. Data information: (C, E, F) Data are presented as mean ± SEM. ns not significant; *$P < 0.05$; **$P < 0.01$ in unpaired two-tailed $t$ test. Source data are available online for this figure.

# Methods

## Reagents and tools table

| Reagent/resource | Reference or source | Identifier or catalog number |
|---|---|---|
| **Experimental models** | | |
| Human brain tissue samples | Department of Neuropathology of the Amsterdam UMC | NA |
| HEK293 | ATCC | CRL-1573 |
| BV2 | Przanowski et al, 2018 | NA |
| GL261 luc + /tdT+ | Ochocka et al, 2021 | NA |
| GL261 WT | Ochocka et al, 2021 | NA |
| C57BL/6J (*M. musculus*) | Nencki Institute of Experimental Biology, PAS | NA |
| SorLA−/− (SorLA-KO/SLKO) C57BL/6J mice | Andersen et al, 2005 | NA |
| **Recombinant DNA** | | |
| pEGFPC2-BIO | Swiech et al, 2011 | NA |
| GFP-TNFα | Manderson et al, 2007 | Addgene #28089 |
| SorLA-ΔVPS10P | This study | NA |
| SorLA-ΔEGF/β-propeller | This study | NA |
| SorLA-ΔFN3 | This study | NA |
| SorLA-ΔCR | Mehmedbasic et al, 2015 | NA |
| SorLA-WT | This study | NA |
| SorLA mini-VPS10P | This study | NA |
| SorLA mini-EGF/β-propeller | This study | NA |
| SorLA mini-FN3 | This study | NA |
| SorLA mini-CR | This study | NA |
| **Antibodies** | | |
| Anti-Caspase-3 | Cell Signaling | CS9662 |
| Anti-CD8-alpha | Abcam | ab217344 |

| Reagent/resource | Reference or source | Identifier or catalog number |
|---|---|---|
| Anti-Galectin-3 | Biolegend | M3/38 |
| Anti-GAPDH | Millipore | MAB374 |
| Anti-GFP | Santa Cruz Biotechnology | SC8334 |
| Anti-GM130 | BD Biosciences | BD610823 |
| Anti-GPX4 | Abcam | ab125066 |
| Anti-Iba1 | WAKO | 019-19741 |
| Anti-Iba1 (for iMG staining) | Abcam | ab5076 |
| Anti-Lamp1 | Sigma-Aldrich | MABC39 |
| Anti-MPO | R&D Systems | AF3667 |
| Anti-myc-tag | Cell Signaling | 2278S |
| Anti-P2RY12 | Genetex | GTX54796 |
| Anti-PARP | Cell Signaling | CS9542 |
| Anti-Rab7 | Cell Signaling | CS95746 |
| Anti-p-RIP1 | Cell Signaling | CS83613 |
| Anti-RIP1 | Cell Signaling | CS3493 |
| Anti-p-RIP3 | Abcam | ab222320 |
| Anti-RIP3 | Abcam | ab62344 |
| Anti-p-STAT3 | Cell Signaling | CS9145 |
| Anti-STAT3 | Cell Signaling | CS9145 |
| Anti-Rab11 | BD Biosciences | BD610657 |
| Anti-SorLA C-term, produced in rabbit | Schmidt et al, 2007 | NA |
| Anti-SorLA | BD Transduction Laboratories | 611861 |
| Anti-SorLA | EMD Millipore | MABN1793 |
| Anti-SorLA, produced in goat | Schmidt et al, 2016 | NA |
| Anti-Tmem119 | Synaptic Systems | 400002 |
| Anti-TNFα | Cell Signaling | CS11948S |
| Anti-TRFR | Abcam | ab269513 |
| Anti-Vti1b | BD Biosciences | BD611404 |
| **Oligonucleotides and other sequence-based reagents** | | |
| Sorl1 TaqMan probe | ThermoFisher Scientific | Hs00983770; Mm01169526 |
| TNFα TaqMan probe | ThermoFisher Scientific | Mm00443258 |

| Reagent/resource | Reference or source | Identifier or catalog number |
|---|---|---|
| β2M TaqMan probe | ThermoFisher Scientific | Hs00187847; Mm00437762 |
| Hprt1 TaqMan probe | ThermoFisher Scientific | Mm00446968; Hs02800695m1 |
| SOX2 TaqMan probe | ThermoFisher Scientific | Hs01053049_s1 |
| P2RY12 TaqMan probe | ThermoFisher Scientific | Hs01881698 |
| TREM2 TaqMan probe | ThermoFisher Scientific | Hs00219132m1 |
| AIF1 TaqMan probe | ThermoFisher Scientific | Hs00610419 |
| GAPDH TaqMan probe | ThermoFisher Scientific | Hs02758991 |
| TBP TaqMan probe | ThermoFisher Scientific | Hs00427620m1 |
| NANOG TaqMan probe | ThermoFisher Scientific | Hs02387400_g1 |
| OCT4 TaqMan probe | ThermoFisher Scientific | Hs00999632_g1 |
| CX3CR1 TaqMan probe | ThermoFisher Scientific | Hs01922583s1 |
| ITGAM TaqMan probe | ThermoFisher Scientific | Hs00167304 |
| gRNA 5´CAGTAGCGTTCGCCCGAACA´3 | This study | NA |
| **Chemicals, enzymes, and other reagents** | | |
| Fetal bovine serum (FBS) | Gibco | #10500064 |
| FBS for microglial culture | PanBiotech | #P30-3302 |
| Dulbecco's modified essential medium (DMEM) | Gibco | #31885 |
| DMEM w/Glutamax | Gibco | #10569-010 |
| DMEM/F12 | Gibco | #11039-021 |
| Insulin-transferrin-selenite | ThermoFisher Scientific | #41400045 |
| B27 | ThermoFisher Scientific | #17504001 |
| N2 | ThermoFisher Scientific | #17502048 |
| Glutamax | ThermoFisher Scientific | #35050038 |
| MEM non-essential amino acids | ThermoFisher Scientific | #11140035 |
| Monothioglycerol | Sigma | #M1753 |
| Insulin | PromoCell | #C-52310 |
| IL-34 | Peprotech | #200-34 |
| TGFβ1 | Peprotech | #100-21C |
| M-CSF | Peprotech | #300-25 |
| CD200 | Novoprotein | #C311 |
| CX3CL1 | Peprotech | #300-31 |
| TNFα | Gibco | PMC3014 |

| Reagent/resource | Reference or source | Identifier or catalog number |
|---|---|---|
| Trypsin | Sigma | T8003 |
| DNAse | Sigma | DN25 |
| Essential 8™ Flex Medium | Gibco | #A2858501 |
| ReLeSR™ | Stem Cell Technologies | #05872 |
| Penicillin/streptomycin solution | Sigma | #P4333 |
| G418 solution | Invivogen | #ant-gn-5 |
| Poly-L-lysine (PLL) | Sigma | #P1274 |
| Lipopolysaccharide (LPS) | Sigma | #L7770, #L4391 |
| Phorbol myristate acetate (PMA) | Sigma | #P8139 |
| Hank's Buffered Saline Solution (HBSS) without $Ca^{2+}$ and $Mg^{2+}$ ions | Gibco | 14175-095 |
| Lipofectamine 2000 reagent | ThermoFisher Scientific | #11668019 |
| D-luciferin sodium salt | Synchem | 103404-75-7 |
| Isoflurane | Polypharm | 5468 |
| Bovine serum albumin (BSA) | Sigma | A9647 |
| Normal donkey serum (NDS) | Sigma | D9663 |
| DAKO fluorescence mounting medium | Agilent | S3023 |
| Antibody diluent | VWR International | #ABB999 |
| Sodium citrate buffer | VWR International | #1.06448.1000 |
| Vectashield | Vector Laboratories | H-1800-10 |
| Nondient P40 (NP40) substitute | Sigma | #11754599001 |
| **Software** | | |
| scTools | This study | https://github.com/MateuszJakiel/scTools |
| Seurat | Hao et al, 2021 | NA |
| MCFS-ID | Dramiński and Koronacki, 2018; Dramiński et al, 2008 | https://cran.r-project.org/web/packages/rmcfs/ |
| Source code for SORL1 expression analysis | This study | https://github.com/mdraminski/expressionLevelsSorLA |
| **Other** | | |
| STEMdiff™ Hematopoietic Kit | Stem Cell Technologies | #05310 |
| High-Capacity RNA-to-cDNA™ Kit | ThermoFisher Scientific | 4387406 |
| ICAM-1 ELISA Kit | Bio-Techne | MIC100 |
| Murine U-Plex Biomarker Group 1 assay | Meso Scale Diagnostics | #K15083K |
| Human TNFa U-PLEX assay | Meso Scale Diagnostics | #K151UCK |
| TaqMan™ Fast Universal PCR Master Mix | ThermoFisher Scientific | 4352042 |

| Reagent/resource | Reference or source | Identifier or catalog number |
|---|---|---|
| Pierce™ BCA Protein Assay Kit | ThermoFisher Scientific | 23225 |
| RNeasy Mini Kit | Qiagen | 74104 |
| Direct-zolTM RNA MiniPrep Kit | Zymo Research | R2053 |
| X-treme in vivo bioluminescence imaging system | Bruker | NA |
| GFP-Trap Magnetic Particles | Chromotek | #gtd |
| Pre-coated glass slides | VWR International | #KNITVS11274077FEA |
| Matrigel | Corning | #356234, #354277 |
| Cell culture inserts with 0.4-µM pores | Falcon | #353090 |
| EDTA tubes | Profilab | #320 |

## Animals

Mice with a targeted disruption of murine SorLA (SorLA-KO) have been described (Andersen et al, 2005). We used SorLA-KO mice on an inbred C57BL/6J background. For primary cultures, SorLA-KO and C57BL/6J wild-type newborns of both sexes were used. For in vivo studies, SorLA-KO and C57BL/6J wild-type males at 10–16 weeks of age were used. Animals were kept under a 12 h/12 h light/dark cycle with free access to food and water. Animal experimentation was performed following approval by the First Ethical Committee in Warsaw (approvals no 1102/2020, 1274/2022) and in agreement with the ARRIVE guidelines.

## Primary microglial cultures

Primary microglial cultures were prepared using whole brains of P0-P1 C57BL/6J wild-type or SorLA-KO newborns. Brains were stripped of meninges and washed three times with HBSS. Afterward, brains were enzymatically dissociated with 1% Trypsin and 0.05% DNAse. Cell culture medium composed of DMEM w/ Glutamax, 10% FBS (PanBiotech), and penicillin/streptomycin was used to inhibit enzymatic digestion. Next, tissues were centrifuged ($130 \times g$, 4 °C, 10 min), and obtained pellets were resuspended in fresh cell culture medium. Cells were plated onto flasks coated with 0.1 mg/mL PLL. After 48 h, cells were washed three times with PBS to remove debris and maintained for the next 8 days. Then, microglial cells were separated from other glial cells by 1 h shaking at 80 RPM at 37 °C, collected by centrifugation ($130 \times g$, 4 °C, 10 min), and seeded onto six-well ($6 \times 10^5$ cells) or 24-well ($1 \times 10^5$ cells) plates, on coverslips. Forty-eight hours after plating microglia were used for PMA and LPS simulations or co-cultured with GL261 cells.

## Cell lines

All cell lines were routinely tested for mycoplasma contamination. HEK293 and GL261 cells were cultured in DMEM with 10% FBS (Gibco), penicillin/streptomycin and with the addition of 100 µg/mL G418, a selection antibiotic, in case of GL261-tdTomato$^+$Luc$^+$.

BV2 cells were cultured in DMEM w/Glutamax with 2% FBS (Gibco) and penicillin/streptomycin. HEK293 transfections were performed with Lipofectamine 2000 reagent, according to the manufacturer's protocol. The cells co-expressing SorLA mutants and TNFα-GFP (or GFP) were collected 24 h later.

## Human tissue samples

Human brain tissue samples used in this study were obtained from the archives of the Department of Neuropathology of the Amsterdam UMC (University of Amsterdam, the Netherlands). Appendix Table S1 summarizes the clinical characteristics of patients. All human specimens were obtained and used in accordance with the Declaration of Helsinki and the Amsterdam UMC Research Code provided by the Medical Ethics Committee of the AMC.

## iPSC culture and differentiation to microglia cells

The human-induced pluripotent stem cell (iPSC) line HMGU001-A/BIHi043-A (WT) and the isogenic SORL1-deficient (SLKO) line BIHi268-A18 line were used in these studies. The SLKO iPSC line was generated by CRISPR/Cas9-mediated genome editing using the gRNA targeting exon 1 in the SORL1 gene (5′-CAG-TAGCGTTCGCCCGAACA-3′). SorLA deficiency in a clone containing an 8 bp frameshift deletion was confirmed by western blotting. iPSC lines were cultured on Matrigel (#356324) coated six-well plates in Essential 8™ Flex Medium (Gibco #A2858501). The culture medium was changed daily, and cells were passaged in clusters every 3–4 days at a density of 80% using 0.5 mM EDTA/PBS. iPSCs were differentiated to microglia using a previously described protocol (McQuade et al, 2018). First, iPSCs were differentiated into hematopoietic progenitors (HPs) using STEM-diff™ Hematopoietic Kit. On day 1, iPSCs with a density of 70–80% were passaged with ReLeSR™ into E8 flex containing Matrigel-coated six-well plates. Clusters at the size of 100 cells were seeded at a density of 50–100 per well. At day 0, in wells with a total of 40–80 clusters, E8 flex medium was replaced with 2 ml of medium A. On day 2, 1 ml medium A was added to the well. On day 3, the medium was removed and replaced with 2 ml of medium B. At days 5, 7, and 9, 1 ml medium B was added to the well. At day 11, HPs were present in the media as well as attached to the bottom. To increase the yield of HPs in the collected media, the adherent cells were gently washed off using a 5-ml serological pipette. The cells were centrifuged at $300 \times g$ for 5 min, resuspended in microglia differentiation medium (DMEM/F12, 2× Insulin-transferrin-sele-nite, 2x B27, 0.5× N2, 1× Glutamax, 1x MEM Non-Essential Amino Acids, 400 µM monothioglycerol, 5 µg/ml Insulin, 100 ng/ml IL-34, 50 ng/ml TGFβ1, and 25 ng/ml M-CSF and seeded at a density of $2 \times 10^5$ cells in Matrigel (#354277) coated six-well plates. At day 13, 15, 17, 19, 21 1 ml of microglia differentiation medium was added to the well. At day 23, the medium, except 1 ml, was transferred to a 15-ml falcon tube to spin down the floating cells at $300 \times g$ for 5 min. The cells were resuspended in 1 ml medium microglia differentiation medium and returned to the well. This was repeated for days 25–35. At day 35, the cells were resuspended in microglia differentiation medium plus 100 ng/ml CD200 and 100 ng/ml CX3CL1 to further mature the microglia. On day 37, 1 ml of the maturation medium was

added to the well. The microglia cells were seeded for functional studies in the period between days 38–42.

## Microglia stimulations and co-culture with glioma cells

For RNA analysis, primary murine microglia were plated onto glass coverslips on a six-well plate ($6 \times 10^5$/well). After 48 h, cells were stimulated with 100 nM PMA or 100 ng/ml LPS (Sigma #L7770) in DMEM w/Glutamax for 24 h. Then cells were washed twice with cold

PBS and collected according to the RNA isolation kit manufacturer's protocol. For cytokine levels analysis, cell medium was collected, centrifuged to remove cell debris, and frozen at −80 °C until use.

For immunostaining, BV2 cells and primary murine microglia were plated onto glass coverslips ($1 \times 10^5$) on a 24-well plate, and 100 nM PMA was applied 24 h or 48 h after plating, respectively. Twenty-four hours later, cells were fixed with 4% PFA in PBS.

For co-culture experiments, GL261 cells were seeded into the cell culture inserts with 0.4-μm pores ($2.5 \times 10^5$ cells/insert).

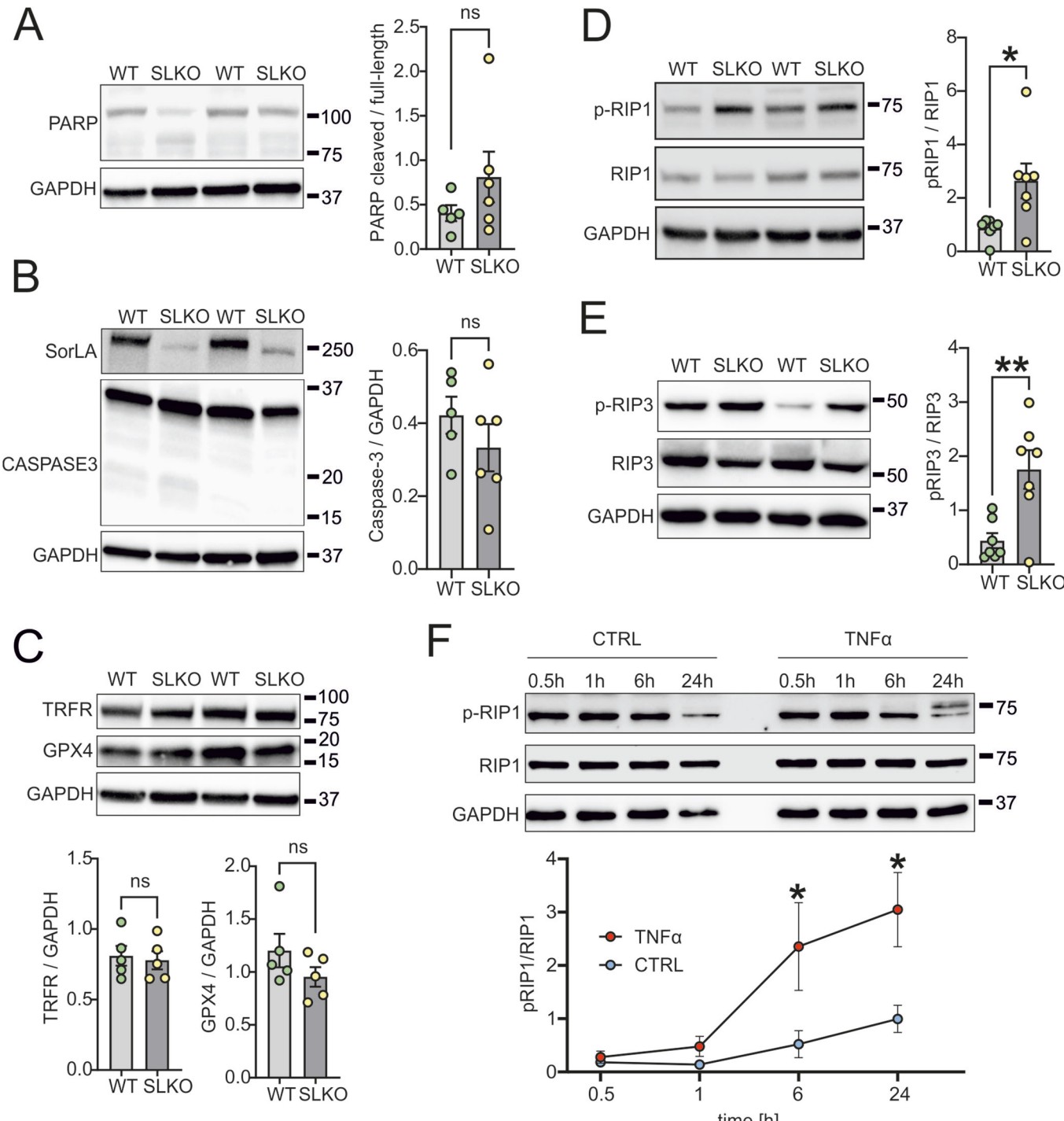

Figure 7. Necroptosis, but not apoptosis or ferroptosis, is activated in gliomas in SorLA-deficient mice.

(A–E) Western blot analysis of apoptosis, ferroptosis and necroptosis markers in WT and SorLA-KO glioma-bearing hemispheres at 21 days post-implantation. Graphs present quantification of the western blot analyses. $n = 5$–7 mice per genotype. (A) Western blot analysis of apoptosis marker PARP which is cleaved upon apoptosis activation to yield bands of lower size. Detection of GAPDH was used as a loading control. Ratio of cleaved and full-length PARP was calculated. (B) Western blot analysis of apoptosis marker caspase-3 which is cleaved upon apoptosis activation. Detection of GAPDH was used as a loading control. Detection of SorLA is also presented. Signal intensity for caspase-3 was normalized to GAPDH. (C) Western blot analysis of ferroptosis markers, TRFR and GPX4. Detection of GAPDH was used as a loading control. Signal intensities for TRFR and GPX4 were normalized to GAPDH. (D) Western blot analysis of necroptosis marker p-RIP1. Detection of RIP1 and GAPDH was used as a control. Signal intensity for p-RIP1 was normalized to RIP1. (E) Western blot analysis of necroptosis marker p-RIP3. Detection of RIP3 and GAPDH was used as a control. Signal intensity for p-RIP3 was normalized to RIP3. (F) Upper panel: western blot analysis of a necroptosis marker p-RIP1 in cultured glioma GL261 cells treated with 20 ng/ml TNFα for indicated time. Cells kept in serum-free medium serve as a control (CTRL). Detection of RIP1 and GAPDH was used as a loading control. Lower panel: quantification of western blot results from three biological replicates. Signal intensity for p-RIP1 was normalized to RIP1. Data information: (A–F) data are presented as mean ± SEM. ns not significant; *$P < 0.05$; **$P < 0.01$ in unpaired two-tailed $t$ test (A–E) or in two-way ANOVA with Sidak's multiple comparisons test (F). Source data are available online for this figure.

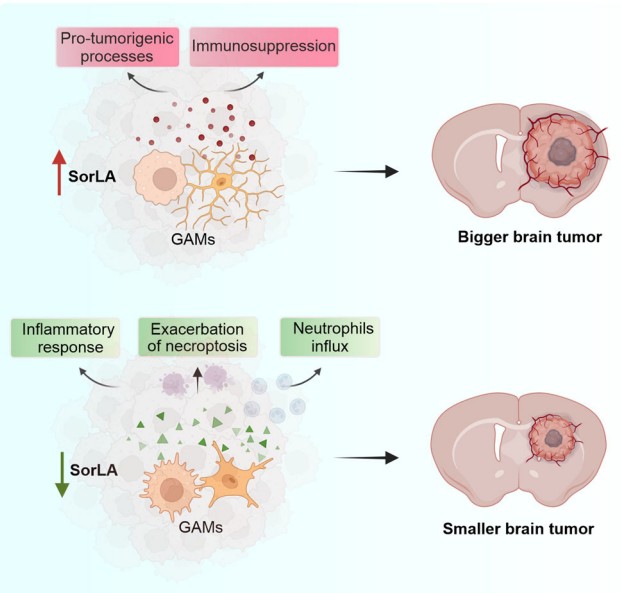

**Figure 8. SorLA has impact on the properties of GAMs, and in consequence on the glioma microenvironment and tumor growth.**

Summary of the main findings. High SorLA levels in GAMs are linked to their pro-tumorigenic properties. Loss of SorLA elicits enhanced inflammatory response, necroptosis and neutrophils influx to the glioma microenvironment, which coincides with inhibition of tumor growth in a murine model of GBM. Created with Biorender.com.

Microglia cells were plated onto glass coverslips on a six-well plate ($6 \times 10^5$/well). After 24 h, GL261 cells were cultured in microglial medium, and after 24 h of medium conditioning over GL261 cells, medium and inserts were transferred to microglia-containing wells. After 24 h of co-culture, microglial RNA was isolated.

iPSC-derived microglia were seeded at a density of $5 \times 10^4$ cells per well in microglia differentiation medium in 96-well plates and stimulated with 1.0 μg/ml LPS (Sigma, #L4391) in microglia differentiation medium. After 24 h, the medium was collected for further analyses and the cells were harvested for RNA isolation.

For TNFα treatment of cultured glioma cells, GL261 cells were plated on a six-well plate ($2.5 \times 10^5$ cells/well) one day prior to the stimulation. Cells were washed once with PBS, and next they were cultured for 0.5, 1, 6, or 24 h in a serum-free medium supplemented with 20 ng/ml TNFα. Cells cultured in a serum-free medium served as a control. Cells were collected at the indicated timepoints and subjected to lysis.

## Immunostaining procedures

Mouse brains were dissected from tumor-bearing mice after transcardial perfusion with PBS and 4% paraformaldehyde PFA in PBS. After post-fixation (PFA, overnight) and cryopreservation in 30% sucrose/PBS, brains were cut in 40-μm free-floating coronal sections using a cryostat. Alternatively, the brains were dissected from non-perfused animals and slowly frozen on isopentane/dry ice. Next, 12-μm coronal sections were cut using a cryostat, immediately mounted on glass slides, and kept at −20 °C until further use. Prior to staining, these sections were thawed at room temperature for 30 min and fixed in 4% PFA/PBS for 30 min. The sections were blocked in 1% horse serum in PBS and incubated with primary followed by secondary antibodies diluted in PBS, supplemented with 1% BSA, 1% NDS, and 0.5% Triton-X. Specimens were washed with PBS, counterstained with DAPI and mounted with DAKO fluorescence mounting medium.

Cells grown on glass coverslips were fixed with 4% PFA/PBS. Next, the cells were washed with PBS, blocked for 1 h in PBS supplemented with 5% NDS and 0.3% Triton-X, and incubated with primary antibodies followed by secondary antibodies diluted in PBS with 1% BSA and 0.3% Triton-X. After washing with PBS and counterstaining with DAPI, the coverslips were mounted with DAKO fluorescence mounting medium.

Human brain tissue was fixed in 10% buffered formalin, embedded in paraffin, sectioned at 5 μm, and mounted on pre-coated glass slides. Sections were deparaffinized in xylene, and ethanol (100%, 95%, 70%). Antigen retrieval was performed using a pressure cooker in 0.01 M sodium citrate buffer (pH 6.0) at 121 °C for 10 min. Slides were washed with PBS (0.1 M, pH 7.4) and incubated overnight with primary antibodies against SorLA (EMD Millipore, 1:150) and Iba1 (1:200) in antibody diluent at 4 °C. The next day, sections were washed with PBS and incubated with Alexa Fluor 568 goat anti-rabbit (1:200) or Alexa Fluor 488 donkey anti-mouse (1:200) antibody plus Hoechst 33258 (1:1000) in antibody diluent for 2 h at room temperature, washed with PBS and mounted with Vectashield.

## Quantitative RT–PCR

Total RNA was extracted from cell lysates and purified with RNeasy Mini Kit or Direct-zolTM RNA MiniPrep Kit, according to the

manufacturer's protocol. Reversely transcribed cDNA was synthesized using High-Capacity RNA-to-cDNA™ Kit. Gene expression was evaluated by quantitative real time-PCR using TaqMan™ Fast Universal PCR Master Mix and the following Taqman Gene Expression Assays: Sorl1, TNFα, β2M, Hprt1, SOX2, P2RY12, TREM2, AIF1, GAPDH, TBP, NANOG, OCT4, CX3CR1, ITGAM. Fold change in gene expression was calculated using the cycle threshold (CT) comparative method (2ddCT) normalizing to CT values for housekeeping genes (in case of primary murine microglia: *β2M* for LPS treatment, *Hprt1* for co-culture with glioma cells and PMA stimulations; in case of human iMG: *GAPDH* for differentiation experiments and *β2M* for LPS treatment; see Appendix Fig. S2).

## ELISA measurements

The following enzyme-linked immunosorbent assay (ELISA) kits were used to determine cytokine levels in microglial media samples and brain tissue lysates of WT and SorLA-KO animals 14 and 21 days after glioma implantation: murine U-Plex Biomarker Group 1 assay, human TNFα U-PLEX assay and ICAM-1/CD54 Quantikine ELISA Kit. All assays were performed according to the manufacturer's protocol.

## Expression plasmids

pEGFPC2-BIO plasmid used for GFP overexpression was a kind gift from Jacek Jaworski (Swiech et al, 2011). GFP-TNFα was a gift from Jennifer Stow (Manderson et al, 2007). The SorLA-ΔCR construct (deletion of Cys1078 to Glu1552) has previously been described (Mehmedbasic et al, 2015). In this study the following constructs were generated: SorLA-ΔVPS10P construct (deletion of Met1 to Pro753) in the pSecTag2A expression vector; SorLA-ΔEGF/β-propeller construct (deletion of Leu754 to Glu1074) in the pcDNA3.1 expression vector; SorLA-ΔFN3 construct (deletion of Val1555-Ala2123) in the pcDNA3.1 expression vector. Human SorLA mini plasmids were generated by amplifying the specific SorLA domain with overlapping primers using PCR technology. The fragments were then ligated into pcDNA3.1 zeo+ vector (Invitrogen). The *SORL1* stop codon was replaced with a myc-coding sequence. All mini-receptors had the original SorLA signal and propeptide sequence and furin cleavage site at the N-terminus, and the original SorLA transmembrane and cytoplasmic domain at the C-terminus. The information on SorLA domains covered by the constructs is provided in Appendix Table S9.

## Co-IP experiments

HEK293 cells overexpressing SorLA mutants (depleted from one of the domain or SorLA mini plasmids) and TNFα-GFP (or GFP) were scraped in lysis buffer containing 10 mM Tris pH 7.5, 150 mM NaCl, 1 mM $CaCl_2$, 1 mM $MgCl_2$, and 0.5% NP40 substitute with proteases and phosphatases inhibitors. The lysate was kept on ice for 30 min and centrifuged ($16,000 \times g$, 4 °C, 15 min). Lysis buffer w/o NP40 substitute was added to the cleared lysate to decrease concentration of the detergent in samples to 0.2%. Protein level was measured by Pierce™ BCA Protein Assay Kit. Part of the resultant cleared lysate was kept for analyses, while the rest (500–600 µg) was used for 30 min incubation with GFP-Trap Magnetic Particles®

(with rotation, in 4 °C) to pull down the GFP tag. Lysis buffer with reduced NP40 substitute content (0.2%) was used for three subsequent washing of beads. Proteins bound to the beads were released by boiling in 2× Laemmli Buffer for 10 min and further examined by western blot.

## Stereotaxic implantation of glioma cells

Implantations were performed as described (Kaminska et al, 2024). In brief, mice were kept under constant anesthesia during the whole procedure using 2% Isoflurane in oxygen. After subcutaneous administration of butorphanol (2 mg/kg body weight) and meloxicam (2 mg/kg body weight), as well as bupivacaine local application (8 mg/kg body weight), skin on the head was incised. The hole in the skull was drilled at the following coordinates: −1 mm anterior-posterior (AP) and 2 mm medial–lateral (ML) from bregma. In total, $8 \times 10^5$ of GL261-tdTomato+Luc+ cells in 1 µl of DMEM were stereotactically injected into the right striatum at the rate of 0.25 µl/min, to a depth of 3 mm according to the brain surface. Withdrawing of the syringe was performed at the rate 1 mm/min to avoid backward outflow of the cell suspension. The skin incision was closed using sutures. After surgery, animals were carefully monitored until they fully recovered from anesthesia. For the following 3 days, mice received meloxicam (2 mg/kg body weight) once a day. Mice weight and well-being were controlled every 2–3 days until the end of the experiment.

## Bioluminescence imaging of glioma growth

Mice implanted with GL261-tdTomato+Luc+ were injected intraperitoneally with D-Luciferin sodium salt (150 mg/kg body weight) in PBS. After 8 min, animals were anesthetized with 2% Isoflurane in oxygen and placed immediately in X-treme Imaging System under 2% isoflurane/oxygen supply. Ten minutes after D-luciferin administration, bioluminescence emission was determined for the total time of 2 min. X-ray images were acquired following bioluminescence imaging. Tumors were visualized 7, 14, and 21 days post-implantation. Bruker Molecular Imaging (MI) Software was used for signal quantification.

## Blood sample collection

Prior to perfusions, samples of venous blood were collected immediately after incision of the right atrium. EDTA tubes were used for material collection. Hematological analyses were performed by Vetlab® company, Warsaw, Poland.

## Tissue lysates preparation

In all, 14 or 21 days after glioma implantation brains were isolated, hemispheres were dissected and homogenized in 20 mM Tris pH 7.5 150 mM NaCl 2 mM EDTA 2 mM $MgCl_2$, and 10% glycerol, supplemented with protease and phosphatase inhibitors. Lysates were incubated on ice for 20 min and centrifuged ($1000 \times g$, 4 °C, 10 min). Triton-X and NP40 substitute were added to the cleared lysate at final concentration of 0.5% and the samples were incubated for 1 h with rotation in 4 °C. Lysates were centrifuged ($16,000 \times g$, 4 °C, 15 min) and protein concentration in the supernatants was measured using Pierce™ BCA Protein Assay Kit.

## Separation of soluble fraction from brain lysates

Twenty-one days post glioma implantation brains were isolated, tumor-bearing hemispheres were dissected and homogenized in 20 mM Tris-HCl pH 7.5, 2 mM $MgCl_2$, and 0.25 M sucrose, supplemented with protease and phosphatase inhibitors. Lysates were incubated on ice for 20 min and centrifuged ($1000 \times g$, 4 °C, 10 min). Part of the resultant cleared lysate was kept for analyses, while the rest was further centrifuged ($100,000 \times g$, 4 °C, 1 h) to obtain the soluble protein fraction. Protein concentration was measured using Pierce™ BCA Protein Assay Kit and samples were subjected to sICAM1 ELISA.

## Western blot

Cell and tissue lysates were analyzed by western blot using the following antibodies: anti-myc-tag (1:1000), anti-SorLA (BD Transduction Laboratories, 1:1000 and C-term homemade 1:1000), anti-GFP (1:250), anti-p-RIP1 (1:1000), anti-RIP1 (1:1000), anti-p-RIP3 (1:1000), anti-RIP3 (1:1000), anti-caspase-3 (1:1000), anti-PARP (1:1000), anti-TRFR (1:1000), anti-GPX4 (1:1000), anti-p-STAT3 (1:2000), anti-STAT3 (1:2000), anti-GAPDH (1:25,000). After incubation with secondary antibodies conjugated to HRP (1:5000), the chemiluminescent signal was registered with the ChemiDoc Imaging System (Bio-Rad).

## Immunostaining

Following antibodies were used: anti-SorLA (1:150, EMD Millipore and 1:100 homemade antibody produced in goat), anti-Iba1 (1:200, WAKO and 1:500, Abcam), anti-P2RY12 (1:100), anti-TNFα (1:200), anti-Vti1b (1:100), anti-Rab11 (1:200), anti-GM130 (1:1000), anti-Lamp1 (1:500), anti-Rab7 (1:100), anti-Tmem119 (1:500), anti-MPO (1:40), anti-galectin-3 (1:500), anti-CD8 alpha (1:500).

## Microglia morphology analysis

For the analysis of microglial cell morphology, Z-stack images of WT and SorLA-KO murine brains after glioma implantation stained for Tmem119 were used. Each cell was selected manually using ImageJ software, and all stacks comprising it were duplicated and saved as a new file for further analyses. 3D reconstruction of microglial branches was performed in Imaris 9.1.2. Software. Imaris Filament Tracer with spot detection mode was used to determine starting and ending points. Automatically added ending points, which did not cover the cell surface were removed manually. Reconstructed cell branches were subjected to Sholl analysis.

## Image quantification

Western blot signals were quantified using the Image Studio Lite software or Image Lab Software 6.1 (Bio-Rad). Microscopy images were quantified using Fiji software. Signal intensities were measured on single-channel images as a mean gray value for the field of view. Colocalization analysis was performed with the Colocalization Threshold plugin with manual selection of the region of interest, containing the cell cytoplasm. Single z-plane images were used for calculating thresholded Mander's coefficient (tM).

## Statistical analysis

For in vivo experiments, an indicated number $n$ is the number of mice per group used in an experiment. For primary cell culture experiments, an indicated number $n$ is the number of independent glial preparations (biological replicates) used for ELISA or qRT-PCR analysis. In case of colocalization studies in primary microglia, $n$ is the number of individual cells quantified in a given experiment, and every experiment was replicated at least three times on independent microglial cultures. Statistical analyses were performed using GraphPad Prism software. Where applicable, outlier analysis was performed using ROUT test. Data variance was tested with GraphPad Prism and was similar for the compared groups. The details of statistical analysis are specified in the figure legends. No blinding was performed.

## scRNA-seq data analysis

Initially, scRNA-seq pipeline was established to run analysis on Abdelfattach et al, dataset (Abdelfattah et al, 2022), but to verify obtained results, the pipeline was then adjusted to other scRNA/snRNA-seq datasets (see below). From Abdelfattah et al, only newly diagnosed glioblastoma (ndGBM) samples were selected, consisting of 11 patients (6 males and 5 females), some of whom had multiple samples taken (Abdelfattah et al, 2022). Downstream analysis of the raw count matrices obtained from the GEO series GSE182109 was performed using the R package scTools (1.0) (https://github.com/MateuszJakiel/scTools) with R version 4.2.0. A list of samples was created with scTools function scCreateRawSamples(), from which cells expressing at least 500 genes and genes expressed in at least 5 cells were selected. Then, using the scIntroQC() function the mitochondrial, ribosomal and red blood cell gene percentages were calculated. Only cells with mitochondrial read percentage <15%, ribosomal read percentage > 5% and red blood cell read percentage <0.5% were kept. Cells with gene counts that were less than 3000 were kept. These filtered samples were integrated using the scIntegrate() function. First, the SCTransform (Hafemeister and Satija, 2019) algorithm was applied on all samples. The most informative 3000 genes were selected to perform the Principal Component Analysis (PCA) and the first 30 Principal Components were used for cell clustering and Uniform Manifold Approximation and Projection algorithm (UMAP) applied (McInnes et al, 2020). Markers of obtained clusters were found with scMarkers() that utilizes Seurat's (Hao et al, 2021) FindAllMarkers() function, its criteria were: gene presence in at least 50% of cells, only positive fold change (FC) and the logFC threshold equal to 0.25. Expert biological analysis was used to manually annotate glioma-associated microglia/macrophages cells (GAMs) that were used for further analysis. The integrated counts were extracted from the Seurat Object and were used to perform further calculations. To analyze *SORL1* expression patterns in two independent datasets (ndGBMs from Abdelfattah et al, primary GBM form Neftel et al,) as well as to prepare normalized input matrix to CellChat tool, we used SCTransform method for gene expression normalization and variance stabilization (Hafemeister and Satija, 2019).

Natural Language Processing (NLP) methods were applied to cluster GAMs marker genes and retrieve sets of keywords to describe each cluster. Marker genes descriptions, gathered from various molecular databases provided by BiomaRt (Durinck et al,

2009) such as: NCBI, Gene Ontology, KEGG, Reactome, WikiPathways and Biocarta, were used. To perform this operation each gene was represented as the text document (combined from all available descriptions), and later, to calculate TF-IDF (term frequency–inverse document frequency), as a bag of words/terms. When each gene is represented by a TF-IDF vector it is possible to calculate cosine similarity between the documents and a hierarchical clustering model can be built based on these similarities. Finally, discovering a set of unique keywords that characterize each cluster provides a good functional, biological overview of input genes and groups that should be studied more closely.

The Spearman correlation of *SORL1* expression with other selected genes was calculated using cor.test(); *P* values were adjusted using the FDR method with the threshold equal to 0.05. To detect more complex nonlinear relationships between *SORL1* expression and other genes, the MCFS-ID algorithm from the rmcfs R package (Dramiński and Koronacki, 2018; Dramiński et al, 2008) was applied. *SORL1* gene expression values were discretized by dividing the values of expression into three ranges: {(−3.1374856; −0.3428585], (−0.3428585; 0.8406330], (0.8406330, +inf]} i.e., "low", "medium", and "high" levels. This discretized *SORL1* variable was used as the response in the decision table and all other gene counts were used as explanatory features resulting in 38,086 cells and 3000 genes. Next, 75% of cells from the input decision table were sampled to establish a training set, leaving the remaining cells for the validation set. The significant features set returned by MCFS-ID was obtained using the permutation method and verified its predictive quality on the validation set using the following set of classifiers: decision tree, logistic regression, random forest, Naive Bayes and Support Vector Machines. Finally, to show differential levels of gene expression, in the context of discretized values of *SORL1*, selected top genes from the MCFS-ID returned ranking were visualized on the heatmap using pheatmap R package (Kolde, 2012).

To perform inference, analysis, and visualization of cell–cell communication from single-cell and spatially resolved transcriptomics the R package CellChat (Jin et al, 2023) was used on 21 clusters from ndGBMs (Abdelfattah et al, 2022). These clusters were grouped into seven groups: GAMs [1, 2, 3, 6, 10, 13, 16], lymphocytes [5, 17, 19], tumor cells [0, 7, 8, 9, 15], smooth muscle cells [11], endothelial cells [14], oligodendrocytes [12], and other cells [4, 18, 20]. Moreover, the GAMs group was split into three subgroups based on the corresponding *SORL1* gene expression values: GAMs_low [0; 0.5], GAMs_med (0.5; 1.5], GAMs_hi (1.5, 3] and these bins were based on the result of the *SORL1* histogram (Appendix Fig. S1C).

Other datasets analyzed in this study were GSE163120 (Pombo Antunes et al, 2021), GSE141383 (Chen et al, 2021), GSE131928 (Neftel et al, 2019), GSE135437 (Sankowski et al, 2019), and GSE174554 (Wang et al, 2022b). Data processing and analysis were performed according to the pipeline described above, where relevant.

## Data availability

The source code related to the paper analysis is located in a GitHub repository: https://github.com/mdraminski/expressionLevelsSorLA; https://github.com/MateuszJakiel/scTools; https://cran.r-project.org/web/packages/rmcfs/. Related datasets are provided as Datasets EV1–6. No primary datasets have been generated and deposited. The source data related to Fig. 1A are deposited in the BioStudies database (S-BSST1327).

## Peer review information

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

## Acknowledgements

The authors are indebted to the Stem Cell Facility of the MDC for the generation of the *SORL1*-deficient iPSC line and for providing microglia differentiation protocols. We thank Tatjana Pasternack and Beata Kaza for expert technical assistance. Part of the confocal imaging and 3D reconstructions were performed at the Polish Euro-BioImaging Node 'Advanced Light Microscopy Node Poland'. This work was supported by the project financed by the Minister of Education and Science based on contract No 2022/WK/05, by the Foundation for Polish Science co-financed by the European Union under the European Regional Development Fund (Homing program, POIR.04.04.00 00 5CEF/18 00; to ARM), by the National Science Centre (OPUS program, 2020/37/B/NZ3/00761, to ARM; Preludium program, 2023/49/N/NZ4/01690, to PK), and by the I.3.4 Action of the Excellence Initiative—Research University Programme at the University of Warsaw (to ARM).

## Author contributions

**Paulina Kaminska**: Funding acquisition; Investigation; Visualization; Methodology; Writing—original draft; Writing—review and editing. **Peter L Ovesen**: Investigation; Visualization; Methodology. **Mateusz Jakiel**: Software; Formal analysis; Investigation. **Tomasz Obrebski**: Investigation. **Vanessa Schmidt**: Resources; Supervision; Investigation; Methodology. **Michal Draminski**: Software; Formal analysis; Investigation; Visualization; Methodology. **Aleksandra G Bilska**: Formal analysis; Investigation. **Magdalena Bieniek**: Investigation. **Jasper Anink**: Investigation. **Bohdan Paterczyk**: Investigation. **Anne Mette Gissel Jensen**: Resources. **Sylwia Piatek**: Investigation. **Olav M Andersen**: Resources; Supervision. **Eleonora Aronica**: Supervision. **Thomas E Willnow**: Resources; Supervision. **Bozena Kaminska**: Supervision. **Michal J Dabrowski**: Software; Formal analysis; Supervision; Investigation; Visualization; Methodology. **Anna R Malik**: Conceptualization;

Supervision; Funding acquisition; Investigation; Visualization; Methodology; Writing—original draft; Project administration; Writing—review and editing.

## Disclosure and competing interests statement

The authors declare no competing interests.

# Expanded View Figures

**Figure EV1.   *SORL1* is expressed in GAMs.** ▶

(**A**) UMAP projection of single cells from human ndGBM tumors described in Abdelfattah et al, grouped in 21 clusters. Clusters identified as GAMs are indicated with purple dashed line. (**B, C**) UMAP projections presenting expression of *SORL1* (**B**) and GAMs marker genes (**C**) in all clusters as exemplified in (**A**). Expression levels are normalized with SCT. (**D**) UMAP projection of single cells from human GBM tumors described in Neftel et al, grouped in 21 clusters. Cluster identified as GAMs is indicated with purple dashed line. (**E, F**) UMAP projections presenting expression of *SORL1* (**E**) and GAMs marker genes (**F**) in all clusters as exemplified in (**D**). Expression levels are normalized with SCT.

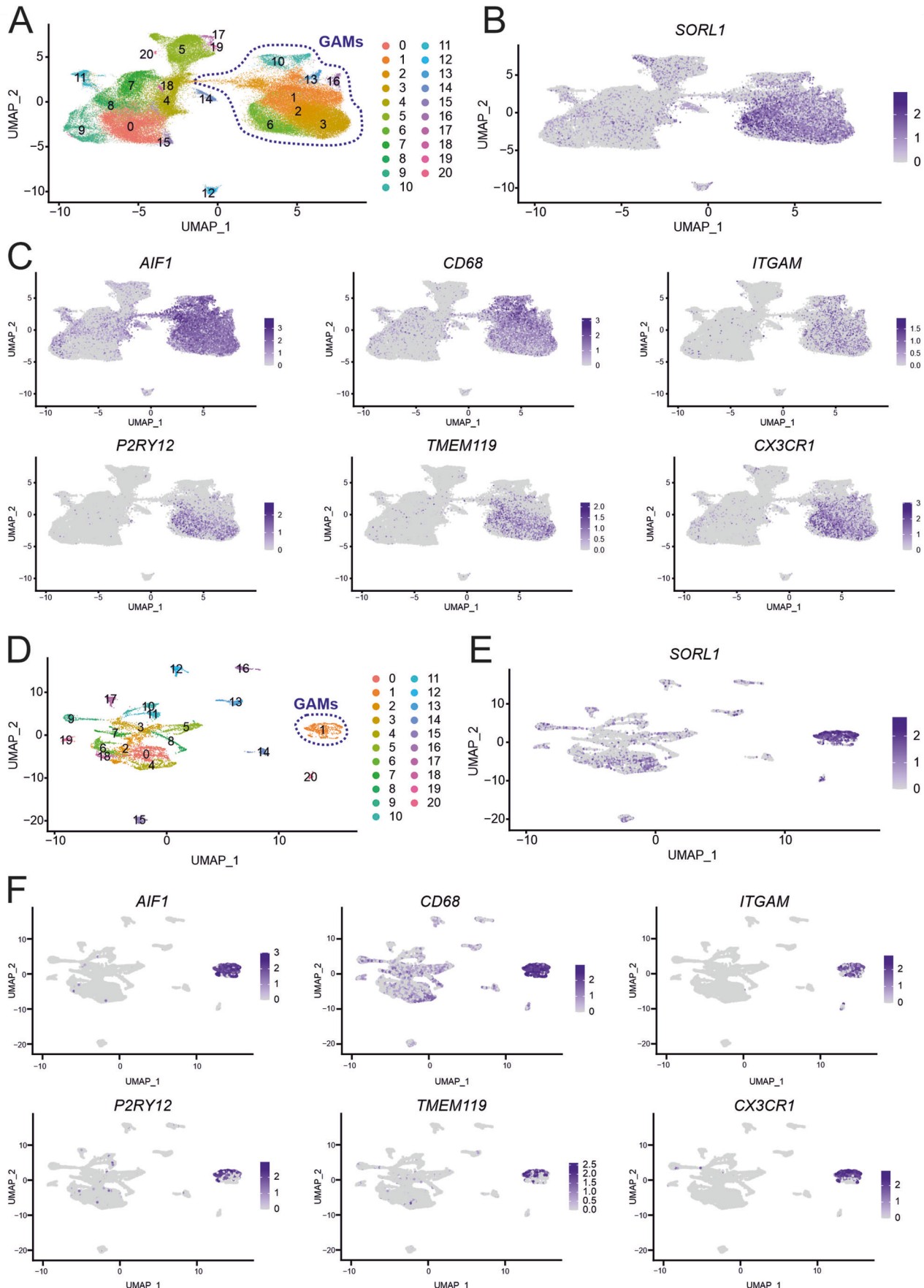

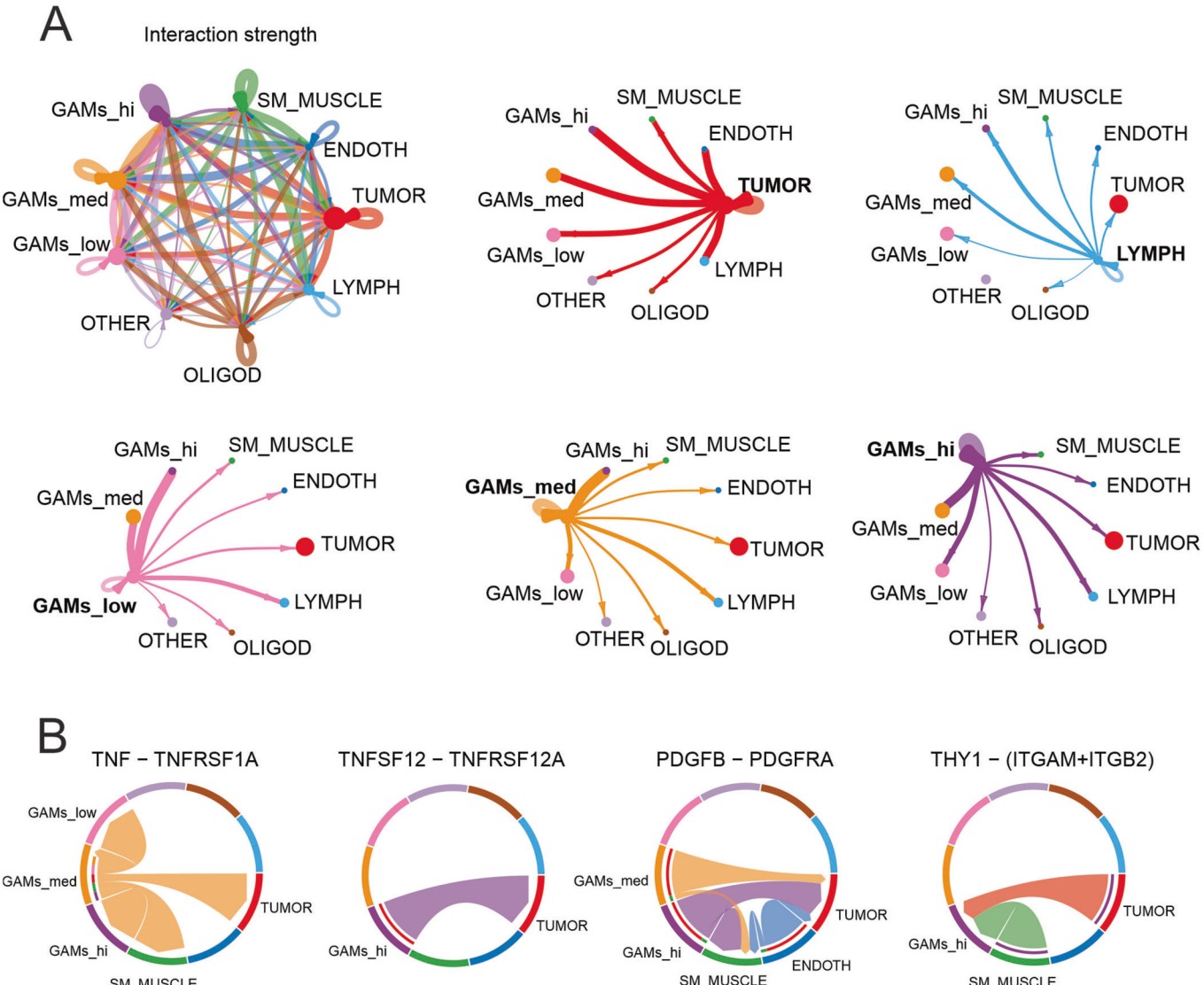

**Figure EV2. Interaction networks between cell populations in newly diagnosed GBM tumors.**

(A) Network plots showing strength of ligand-receptor interactions between cell populations. The line width is proportional to the number of ligand-receptor pairs identified. (B) Chord diagrams indicating selected ligand-receptor pairs mediating interaction between cell populations. Width of chords is proportional to signal strength of the given ligand-receptor pair.

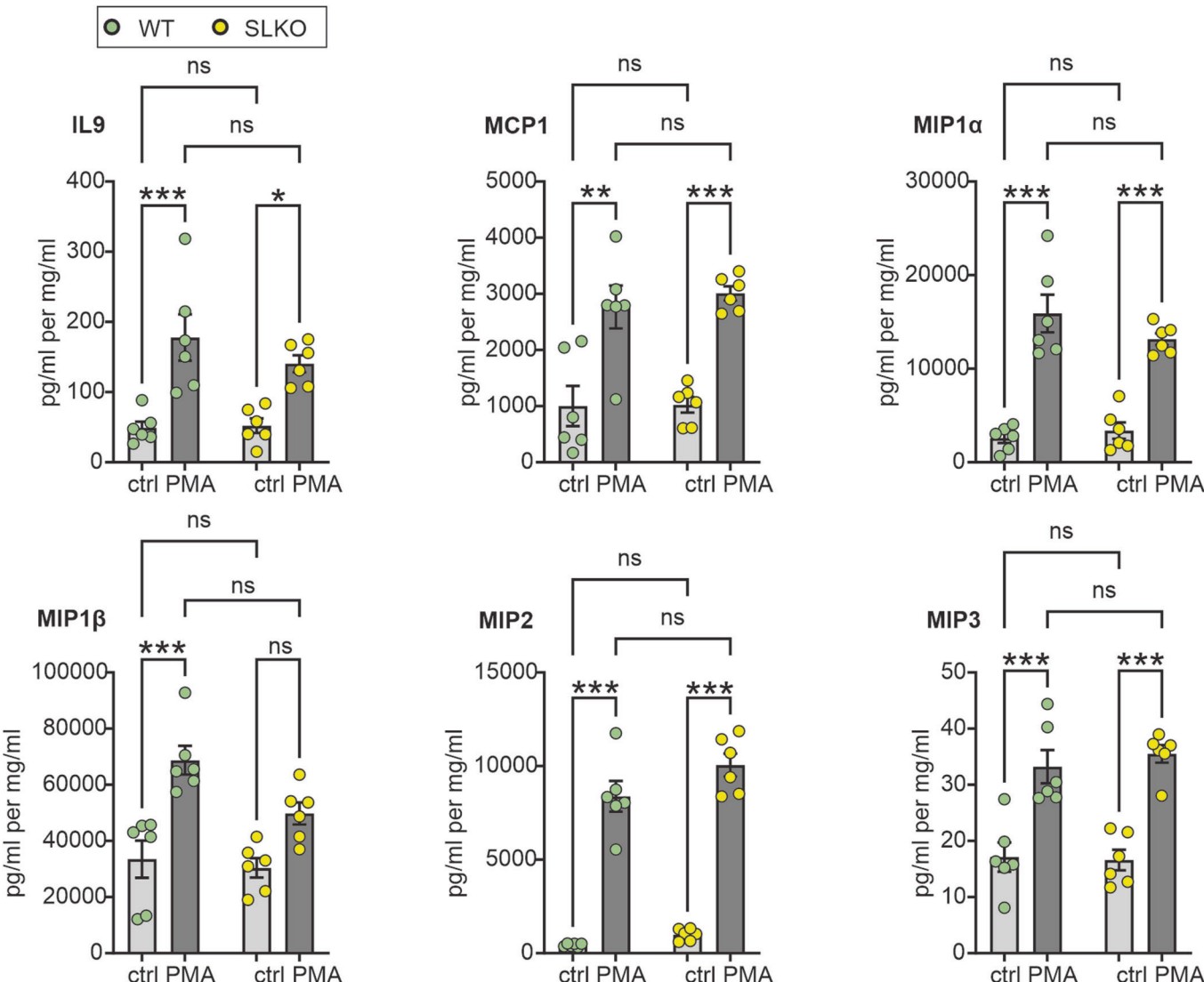

**Figure EV3.  SorLA deficiency has no major impact on microglial cytokine secretion.**

Cytokine levels as determined by ELISA in cell culture medium from primary WT and SorLA-KO microglia either untreated (ctrl) or treated with PMA for 24 h. Cytokine levels were normalized to the protein content in the respective cell lysates. $n = 6$ biological replicates. Data information: data are presented as mean ± SEM. ns not significant; *$P < 0.05$; **$P < 0.01$; ***$P < 0.001$ in two-way ANOVA with Tukey's multiple comparisons test.

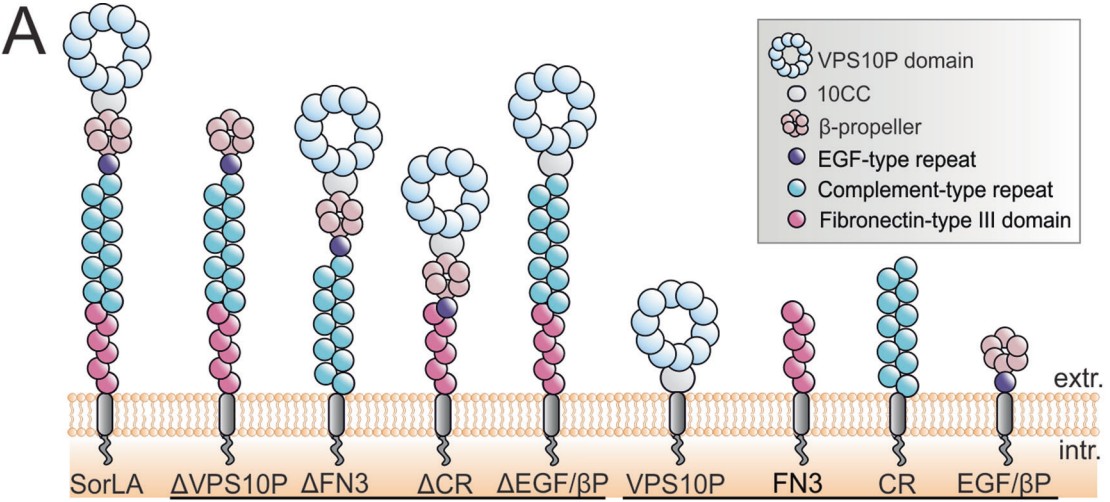

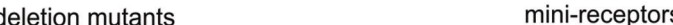

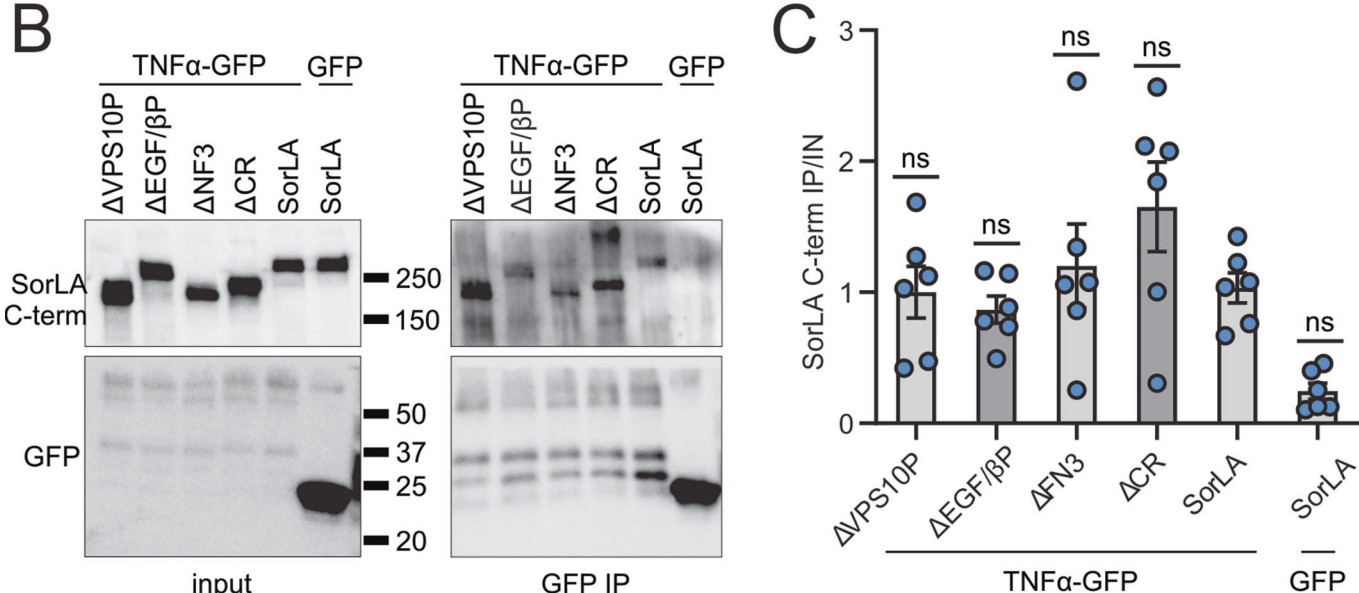

**Figure EV4.** Binding of SorLA mutants to TNFα.

(A) Schematic representation of SorLA structure and the mutant proteins used in this study. (B) Co-immunoprecipitation (co-IP) of SorLA deletion mutants with TNFα-GFP overexpressed in HEK293 cells after GFP-IP. GFP serves as a negative control. SorLA was detected using an antibody raised against its C-terminus. (C) Quantification of the results of 6 biological replicates as exemplified in (B). Ratio of co-IP and input signals (IP/IN) was calculated for each transfection variant. Data information: (C) Data are presented as mean ± SEM. ns not significant in one-way ANOVA with Tukey's multiple comparisons test, comparing to full-length SorLA.

