## [Peer Review File · EMBO Reports]

SorLA restricts TNF α release from microglia to shape glioma-supportive brain microenvironment

Paulina Kaminska, Peter Ovesen, Mateusz Jakiel, Tomasz Obrebski, Vanessa Schmidt, Michal Draminski, Aleksandra Bilka, Magdalena Bieniek, Jasper Anink, Bohdan Paterczyk, Anne Mette G. Jensen, Sylwia Piatek, Olav Andersen, Eleonora Aronica, Thomas Willnow, Bozena Kaminska, Michal Dabrowski, and Anna Malik

Corresponding author(s): Anna Malik (ar.malik@uw.edu.pl)

Review Timeline:

Submission Date:	30th Aug 23
Editorial Decision:	10th Oct 23
Revision Received:	30th Jan 24
Editorial Decision:	19th Feb 24
Revision Received:	25th Feb 24
Accepted:	1st Mar 24

Editor: Achim Breiling

Transaction Report:

Dear Dr. Malik,

Thank you for the submission of your research manuscript to EMBO reports. I have now received the reports from the three referees that were asked to evaluate your study, which can be found at the end of this email.

As you will see, the referees think that the findings are of high interest. However, the referees have several comments, concerns, and suggestions, indicating that a major revision of the manuscript is necessary to allow publication of the study in EMBO reports. As the reports are below, and all the referee concerns need to be addressed, I will not detail them here.

Given the constructive referee comments, I would like to invite you to revise your manuscript with the understanding that all referee concerns must be addressed in the revised manuscript and in a detailed point-by-point response. Acceptance of your manuscript will depend on a positive outcome of a second round of review. It is EMBO reports policy to allow a single round of revision only and acceptance of the manuscript will therefore depend on the completeness of your responses included in the next, final version of the manuscript.

1) a .docx formatted version of the final manuscript text (including legends for main figures, EV figures and tables), but without the figures included. Figure legends should be compiled at the end of the manuscript text.

2) individual production quality figure files as .eps, .tif, .jpg (one file per figure), of main figures (up to 8) and EV figures. Please upload these as separate, individual files upon re-submission.

4) a complete author checklist, which you can download from our author guidelines

(<https://www.embopress.org/page/journal/14693178/authorguide>). Please insert page numbers in the checklist to indicate where the requested information can be found in the manuscript. The completed author checklist will also be part of the RPF.

5) that primary datasets produced in this study (e.g. RNA-seq, ChIP-seq, structural and array data) are deposited in an

appropriate public database. If no primary datasets have been deposited, please also state this in a dedicated section (e.g. 'No primary datasets have been generated and deposited'), see below.

The accession numbers and database should be listed in a formal "Data Availability" section (placed after Materials & Methods) that follows the model below. This is now mandatory (like the COI statement). Please note that the Data Availability Section is restricted to new primary data that are part of this study. This section is mandatory. As indicated above, if no primary datasets have been deposited, please state this in this section

Data availability

8) Regarding data quantification and statistics, please make sure that the number "n" for how many independent experiments were performed, their nature (biological versus technical replicates), the bars and error bars (e.g. SEM, SD) and the test used to calculate p-values is indicated in the respective figure legends (also for potential EV figures and all those in the final Appendix). Please also check that all the p-values are explained in the legend, and that these fit to those shown in the figure. Please provide statistical testing where applicable. Please avoid the phrase 'independent experiment', but clearly state if these were biological or technical replicates. Please also indicate (e.g. with n.s.) if testing was performed, but the differences are not significant. In case n=2, please show the data as separate datapoints without error bars and statistics. See also: <http://www.embopress.org/page/journal/14693178/authorguide#statisticalanalysis>

9) Please add scale bars of similar style and thickness to all the microscopic images, using clearly visible black or white bars (depending on the background). Please place these in the lower right corner of the images themselves. Please do not write on or near the bars in the image but define the size in the respective figure legend.

10) Please also note our reference format:

12) We now use CRedit to specify the contributions of each author in the journal submission system. CRedit replaces the author contribution section. Please use the free text box to provide more detailed descriptions and do not provide your final manuscript text file with an author contributions section. See also our guide to authors: <https://www.embopress.org/page/journal/14693178/authorguide#authorshipguidelines>

13) We would encourage you to use 'Structured Methods', our new Materials and Methods format. According to this format, the

Materials and Methods section should include a Reagents and Tools Table (listing key reagents, experimental models, software and relevant equipment and including their sources and relevant identifiers) followed by a Methods and Protocols section in which we encourage the authors to describe their methods using a step-by-step protocol format with bullet points, to facilitate the adoption of the methodologies across labs. More information on how to adhere to this format as well as downloadable templates (.doc or .xls) for the Reagents and Tools Table can be found in our author guidelines (section 'Structured Methods'):

14) Please order the manuscript sections like this, using these names:

Title page - Abstract - Keywords - Introduction - Results - Discussion - Materials and Methods - Data availability section - Acknowledgements - Disclosure and Competing Interests Statement - References - Figure legends - Expanded View Figure legends

I look forward to seeing a revised version of your manuscript when it is ready. Please let me know if you have questions or comments regarding the revision.

Yours sincerely,

Referee #1:

In this study the authors report that SorLA expression in GAMs affects the functional phenotype of these cells. While high expression of SorLA is associated with a pro-tumorigenic function of GAMs, loss of SorLA results in a pro-inflammatory response of GAMS which suppressed tumor growth in a murine model of glioma. Mechanistically, the authors propose that SorLA acts a sorting receptor for TNF α to limit its release from microglia. Overall, the authors report a novel function of SorLA in glioma associated microglia/macrophages, an intracellular shuttle transport previously reported in neurons. While some of the results are well presented I consider that other conclusions should be further validated.

1. The authors showed that the expression of SorLA in GAMs is not found in all GBM patients but actually looking at the only confocal image presented in the manuscript, even not all GAMs express SorLA, so what is the percentage of Iba1+ cells in the tumor that co-express SorLA? Is there a correlation between patient prognosis/survival and the percentage of Iba1+/Sor1- ratio? (if not from the samples stained, which is a very small group of samples, can this information be inferred from the scRNAseq?).
2. Since not many GBM samples showed SorLA expression, I wonder why the authors did not try to validate the expression of Sor1 in GAMs using a different cohort of patients. There are several scRNAseq data sets available from different published studies, and it will further strength the results if using a similar definition for the clusters defined as GAMs in their dataset from Abdelfattah et al, 2022. showed the same results using a different dataset.
3. In the study the authors associate the expression of LGALS1 in pro-inflammatory GAM phenotype, while many studies in the literature have reported the expression of this gene in pro-tumorigenic TAMs inducing an immunosuppressive phenotype. "LGALS1, the gene encoding Gal-1, promotes M2 TAMs secretion of IL-10 and TGF β , which contribute to a tumor-supporting and immunosuppressive environment". Furthermore, high expression of LGALS1 in high grade glioma significantly correlates with poor prognosis. Can the authors discuss/comment this discrepancy?
4. Fig 2E in addition to microscopy images the authors should show some staining with microglia markers to confirm phenotype (not just morphology). For example fluorescent staining and confocal microscopy analysis with P2RY12 and TREM2 markers.
5. Images of co-localization of TNF-a are representative and difficult to assess the actual co-localization... the authors should show graphs of co-localization analysis for several images taken in Appendix Suppl S2 as they did in figure 3G.
6. Figure 4E-F - While using GAPDH as loading control is appropriate, in order to assess changes in phosphorylation in STAT3 it should be compared to total STAT3 protein. Same with p-RIP3 the total RIP3 protein control is missing.
7. The authors show an increase in MPO positive signaling that they associate with high infiltration of neutrophils. Based on the tumor size, these are well established tumors, not early stages of tumor development. And so far, most of the studies in glioma (murine and human) showed a pro-tumorigenic function of tumor associated neutrophils (TANs) at the more advanced stages. I don't think the authors had a chance to read the latest study published in Cell by the Joyce's lab (Maas RR, et al Cell 2023), where they actually showed that one of the contributors to the phenotypic and functional alterations of peripheral blood neutrophils to pro-tumorigenic TANs in GBM and brain metastasis is TNF-alpha. These new results in part contradict the speculations of the authors that high levels of TNF-alpha in SorLA-KO mice promote the anti-tumorigenic function of the recruited neutrophils.

Referee #2:

The manuscript by Kaminska et al. suggests a role of the sorting receptor SorLA in regulating glioma-associated microglia and macrophages (GAMs) function. The authors find altered SorL1 expression in activated microglia. Their data suggest that SorLA deficiency leads to increased release of the pro-inflammatory cytokine TNFalpha from microglia, SorLA interacts with TNFalpha and plays a role in its secretion. Glioma growth is limited in SorLA deficient mice and pro-tumorigenic activities reduced. They observe a neutrophil influx that is likely TNFalpha driven. Finally, they provide evidence of TNF receptor1 mediated necroptosis that might contribute to the elimination of glioma cells and limit tumor growth under SorLA deficiency.

This is an interesting study reporting novel observations. Functional relevance of SorLA in GAM regulation is demonstrated and for most conclusions, sufficient evidence is drawn.

However, there are a number of concerns need be considered to improve the study and its presentation.

1. Although the manuscript is well written, it should be improved, particularly the discussion. The reader would benefit from a graphical model integrating the findings. Using this model for navigation, the discussion should be rewritten.

2. Fig.2: Several information is not presented in a uniform format or remains ambiguous. In this respect the manuscript appears preliminary and deserves improvement.

Examples: Labeling in 2D "FC vs WT Ctrl".

Statistical significance is not uniformly shown, compare e.g. 2A, F and the other subfigures.

Different font size used.

Please carefully revise.

3. Quantitative PCR experiments: Sorl1 mRNA levels were quantified relative to Hprt1 (co-cultivation with glioma) or beta2M (LPS or PMA treatment). Why were different genes used for normalization? How was their suitability as reference genes tested? According to the MIQE guidelines (Bustin et al. 2009) more than one reference gene should be used for each gene of interest.

4. Colocalization (Fig. 3A), should be quantified e.g. by calculating Mander's co-localization coefficient. Based on the current images colocalization is difficult to determine.

5. Co-IP (Fig. 3B), high protein amounts were used in the input, but only low amounts of SorLA precipitated. Co-IP of partial receptors (Fig 3D), results in a high background, notably delta-EGF/betaP is precipitated (3D), but EGF/betaP (3E) seems to mediate the interaction. Together, the presented CoIPs do not convincingly show domain specific interactions. Actually, the current blots suggest that precipitation conditions were not stringent enough.

The authors indicate a "ratio of pull-down and input signals (PD/IN)" which was "calculated for each transfection variant".

Although the given numbers support the authors conclusions, how these have been calculated is difficult to follow. In the methods section the authors mention Western blot quantification. Are the numbers, which are given without statistical analysis, the result of a single experiments? This is not indicated and a great concern.

Clear cut experiments demonstrating the proposed interaction are essential to draw the functional conclusions made in the manuscript.

The study would greatly benefit from a demonstration of direct interaction of SorLA or its specific subdomains and TNFalpha using other, preferably biophysical, methods such as MST or Octet.

6. For colocalization analysis (Fig. 3G) Mander's co-localization coefficient M2 is shown. Please present also M1.

7. Fig. 3G, typo: ttM2 must be tM2

8. Figure legends:

Figure 1C: please indicate as in the results section that the 5 clusters with lowest and highest SorL1 levels are shown.

9. The manuscript could be shortened by providing large parts of figure 1 and 5 as supplemental information.

Referee #3:

In this study, Kaminska and colleagues have explored the role of SorLA in establishing a glioma-supportive brain microenvironment by unraveling its mechanism of action. I believe this study could make a valuable contribution to the field with some additional analyses as follows:

1- The authors utilized data generated by Abdelfattah et al. in 2022 to investigate the gene expression levels of SORL1. While this study includes the expression signature of glioma cells across a vast number of cells, it would be beneficial for the authors to compare the expression signature of their glioma-associated microglia (GAMs) with the expression signature of GAMs identified in other datasets, which are also cited by the authors in the paper (Sankowski et al., 2019). This comparison would

help understand whether the GMA signature is consistent across multiple studies.

2- Although the primary focus of the study is on the specific gene SORL1, performing functional analysis of GAMs using gene ontology tools could provide a more detailed understanding of the functional states of the microglia clusters.

3- To gain a better insight into the microenvironmental state of the tumor cells, the authors could conduct cell-cell interaction analysis using appropriate tools like CellChat or NichNet on the single-cell RNA-seq datasets. These analyses would shed light on the secretory profiles of the cells and, importantly, provide evidence regarding the receptors on the target cells, as the authors have claimed that the "presence of the receptor in host cells is critical for establishing a tumor-promoting microenvironment."

We would like to thank the reviewers for their valuable suggestions and constructive critique. Below, we present our point-by-point response to their comments.

Referee #1:

In this study the authors report that SorLA expression in GAMs affects the functional phenotype of these cells. While high expression of SorLA is associated with a pro-tumorigenic function of GAMs, loss of SorLA results in a pro-inflammatory response of GAMs which suppressed tumor growth in a murine model of glioma. Mechanistically, the authors propose that SorLA acts a sorting receptor for TNF α to limit its release from microglia. Overall, the authors report a novel function of SorLA in glioma associated microglia/macrophages, an intracellular shuttle transport previously reported in neurons. While some of the results are well presented I consider that other conclusions should be further validated.

1. The authors showed that the expression of SorLA in GAMs is not found in all GBM patients but actually looking at the only confocal image presented in the manuscript, even not all GAMs express SorLA, so what is the percentage of Iba1+ cells in the tumor that co-express SorLA? Is there a correlation between patient prognosis/survival and the percentage of Iba1+/SorLA- ratio? (if not from the samples stained, which is a very small group of samples, can this information be inferred from the scRNAseq?).

We agree that variability of SorLA levels in Iba1+ cells in glioma samples is intriguing. To have a closer look at this phenomenon, we now performed new analyses of 6 scRNAseq datasets. As shown in Table S2 in the revised manuscript, the percentage of AIF1+ (*AIF1* encodes for Iba1) cells in tumors that co-express SorLA is variable and ranges from 17.2% to 97% depending on the study. At present we do not know the reasons for this variability. In the dataset that served as the starting point for the majority of bioinformatics analyses in our study (newly-diagnosed GBM from Abdelfattah et al., *Nat Commun.* 2022, PMID: 35140215), 53.1% of AIF+ cells co-expressed SorLA (Table S2).

Unfortunately, to the best of our knowledge, there are no published data in which scRNAseq results are coupled to clinical data, such as patients' survival. In the only publication containing both types of data that we found (Wang et al., *Nat Cancer* 2022, PMID: 36539501), the authors used single-nuclei RNAseq, and not scRNAseq. This might explain why only 0.6% of cells in the dataset were AIF1+ (Table S2 in the revised manuscript). Also, the expression levels of other GAMs markers are relatively low in this dataset in all clusters (see Figure 1A-B for reviewers). Still, we extracted the data on survival of primary GBM patients from this dataset (Figure 1C for reviewers). The total number of patients eligible for survival analysis is 12, and the cell numbers for each of them range from 10 to 3365. Based in these numbers, we concluded that further analysis of this dataset, aiming at evaluating potential correlation between the proportion of SORL1+ cells among AIF1+ cells and patients survival, is not feasible.

In conclusion, although undoubtedly interesting, this type of information cannot be inferred from available data. Thus, at present it is not known whether lower SorLA levels in microglia/macrophages would be beneficial for GBM patients, in spite of the clear effect of

SorLA depletion on tumor size in a murine model of glioma. We have included this remark in the discussion (page 13).

2. Since not many GBM samples showed SorLA expression, I wonder why the authors did not try to validate the expression of Sor1 in GAMs using a different cohort of patients. There are several scRNAseq data sets available from different published studies, and it will further strength the results if using a similar definition for the clusters defined as GAMs in their dataset from Abdelfattah et al, 2022. showed the same results using a different dataset

As suggested, we performed additional analyses of available scRNAseq data. Firstly, as mentioned above, we analyzed several datasets for the proportions of *AIF1*+ cells co-expressing *SORL1*. Using the same pipeline that we used before to define GAMs clusters and to evaluate *SORL1* expression patterns, we also repeated key analyses on additional dataset. To this end, we focused on a study published by Neftel et al. (*Cell* 2019, PMID: 31327527), which contained data for non-immune and immune cells, similarly to the study by Abdelfattah et al. (*Nat Commun.* 2022, PMID: 35140215) (please see Table S2). Although we noted striking differences in the numbers of cells sequenced in both studies (80 157 in ndGBM from Abdelfattah et al.; 7 894 in Neftel et al.) our definition of GAMs clusters performed well in the additional dataset. As in our first analysis, *SORL1* showed the highest expression in GAMs cluster and high vs low *SORL1* expression in GAMs was linked to the properties of GAMs. Overall, our key findings were recapitulated in this dataset which is summarized on Figures EV1, S1, and Table S8.

3. In the study the authors associate the expression of *LGALS1* in pro-inflammatory GAM phenotype, while many studies in the literature have reported the expression of this gene in pro-tumorigenic TAMs inducing an immunosuppressive phenotype.

"*LGALS1*, the gene encoding Gal-1, promotes M2 TAMs secretion of IL-10 and TGF β , which contribute to a tumor-supporting and immunosuppressive environment". Furthermore, high expression of *LGALS1* in high grade glioma significantly correlates with poor prognosis. Can the authors discuss/comment this discrepancy?

Regarding the association of high *LGALS1* expression in high grade glioma with poor prognosis, TCGA and CGGA databases only contain bulk RNA expression data. Therefore, it is impossible to distinguish in which cell type particular genes are expressed (e.g. cancer cells vs GAMs). Thus, data derived from TCGA and CGGA rarely enable direct interpretation of the scRNAseq results. As in case of *SORL1*, it remains an open question whether *LGALS1* levels in microglia/macrophages are associated with patients' survival.

In the first version of the manuscript, we wrote 'we identified *LGALS1*, *S100A10* and pro-inflammatory *MIF1* as predictors of low *SORL1* expression in GAMs'. We did not comment on the relevance of *LGALS1* or *S100A10* levels for the properties of GAMs. Now we have realized that this finding needs to be discussed in the manuscript. Thus, in the revised version, we address potential interpretations of *LGALS1* and *S100A10* levels in GAMs (page 6, first paragraph). In brief, the potential role of galectin-1 in microglia/macrophages seems to be complex. In particular, interpretation of its expression patterns might strongly depend on the cell type analyzed.

To the best of our knowledge, the statement cited by the reviewer ('LGALS1, the gene encoding Gal-1, promotes M2 TAMs secretion of IL-10 and TGF β (...)) refers to the findings by Chen et al. (*Int J Cancer* 2019, PMID: 30613962). In this study, the role of *LGALS1* in tumor cells was investigated. Deleting *LGALS1* in glioma cells modified the composition of glioma-secreted factors. *LGALS1*-deficient glioma cells showed impact on GAMs phenotypes, when implanted to mice. In line with these findings, it was also shown that, when applied to cultured microglia, galectin-1 was able to reduce pro-inflammatory potential of these cells (Starossom et al., *Immunity* 2012, PMID: 22884314).

However, expression of *LGALS1* specifically in microglia/macrophages appears to have different implications. For instance, Kiss et al. (*Brain Res.* 2023, PMID: 37557976) have shown recently that primary microglia stimulated with pro-inflammatory LPS upregulate galectin-1 expression. Also, in the aging brain, there is a subset of *LGALS1* expressing microglia. Some studies have proposed *LGALS1* as a signature gene of microglia subpopulation called PAMs ('proliferative-region-associated microglia'; Li et al., *Neuron* 2019, PMID: 30606613). Finally, *LGALS1* has been shown to be a marker gene for a subpopulation of disease-associated microglia in multiple sclerosis (Masuda et al., *Nature* 2019, PMID: 30760929).

4. Fig 2E in addition to microscopy images the authors should show some staining with microglia markers to confirm phenotype (not just morphology). For example fluorescent staining and confocal microscopy analysis with P2RY12 and TREM2 markers.

Following the reviewer's recommendation, we have improved Figure 2 (Figure 3 in the revised manuscript) and added a new Figure S3 to include more data on microglia differentiation from the iPS cells. Specifically, we have added:

- qPCR data from iPSCs->iMGs differentiation experiments, showing loss of pluripotency markers (*SOX2*, Fig. 3F; *NANOG*, *OCT4*, Fig. S3A) and an increase in expression levels of microglia markers (*P2RY12*, *TREM2*, *AIF1*, Fig. 3F; *CX3CR1*, *ITGAM*, Fig. S3A) during differentiation;

- microscopy images of iMG cells immunostained for microglia markers P2RY12 and Iba1 (Fig. 3G);

- qPCR data showing relative expression of microglia markers *AIF1*, *CD11B*, *P2RY12*, as well as of *SORL1* in WT and SorLA-deficient iMG (Fig. S3B);

- results of western blot analyses detecting SorLA in WT and SorLA-deficient iMG cells, (Figure S3C).

5. Images of co-localization of TNF- α are representative and difficult to assess the actual co-localization... the authors should show graphs of co-localization analysis for several images taken in Appendix Suppl S2 as they did in figure 3G.

This figure has been improved to show more clearly the colocalization of TNF α with the markers of chosen subcellular compartments (Lamp1, Rab1, GM130). In detail, we have provided images of better quality and added insets showing representative cell images in higher magnification. Quantification of the extent of colocalization is also given (Fig. S4 in the revised manuscript). In addition, we also added the information on the antibodies used in the Reagents and Tools Table.

6. Figure 4E-F - While using GAPDH as loading control is appropriate, in order to assess changes in phosphorylation in STAT3 it should be compared to total STAT3 protein. Same with p-RIP3 the total RIP3 protein control is missing.

As suggested, we included new western blot analyses of total STAT3 levels in our glioma samples. Accordingly, we now document quantification of pSTAT3 levels relative to total STAT3 levels, replacing the pSTAT3/GAPDH data (Fig. 5E-F in the revised manuscript).

Regarding the pRIP levels, while re-evaluating the western blot data presented in the manuscript, we realized that the protein was wrongly labeled as pRIP3 as the antibody that was used (Cell Signaling, CS83613, as listed in the Reagents and Tools Table) detects another marker of necroptosis, namely pRIP1. We apologize for this mistake, which has now been corrected. In the revised manuscript, we show the results of western blot analyses of both pRIP1 and pRIP3 (Figure 7D-E). Following the reviewer's suggestion, we have added the results of the western blot analyses of total RIP1 and RIP3 levels and we included respective quantifications.

7. The authors show an increase in MPO positive signaling that they associate with high infiltration of neutrophils. Based on the tumor size, these are well established tumors, not early stages of tumor development. And so far, most of the studies in glioma (murine and human) showed a pro-tumorigenic function of tumor associated neutrophils (TANs) at the more advanced stages. I don't think the authors had a chance to read the latest study published in *Cell* by the Joyce's lab (Maas RR, et al *Cell* 2023), where they actually showed that one of the contributors to the phenotypic and functional alterations of peripheral blood neutrophils to pro-tumorigenic TANs in GBM and brain metastasis is TNF-alpha. These new results in part contradict the speculations of the authors that high levels of TNF-alpha in SorLA-KO mice promote the anti-tumorigenic function of the recruited neutrophils.

Thank you for drawing our attention to this study. Of note, Maas et al. (*Cell* 2023, PMID: 37769657) conclude that 'In the context of inflammation and cancer, TNF α has been associated with induction of either an anti-, or pro-tumoral inflammatory TAN phenotype. Thus, its role in instructing TANs is highly context dependent'. This statement underscores the complexity of functional interactions in the tumor microenvironment. We agree that the role of TNF α and neutrophils in GBM is not fully understood. We have now extended the relevant part of the discussion to provide a more comprehensive view on this topic (page 12, last paragraph).

Referee #2:

The manuscript by Kaminska et al. suggests a role of the sorting receptor SorLA in regulating glioma-associated microglia and macrophages (GAMs) function. The authors find altered SorL1 expression in activated microglia. Their data suggest that SorLA deficiency leads to increased release of the pro-inflammatory cytokine TNFalpha from microglia, SorLA interacts with TNFalpha and plays a role in its secretion. Glioma growth is limited in SorLA deficient mice and pro-tumorigenic activities reduced. They observe a neutrophil influx that is likely TNFalpha driven. Finally, they provide evidence of TNF receptor1 mediated necroptosis that might contribute to the elimination of glioma cells and limit tumor growth under SorLA deficiency.

This is an interesting study reporting novel observations. Functional relevance of SorLA in GAM regulation is demonstrated and for most conclusions, sufficient evidence is drawn.

However, there are a number of concerns need be considered to improve the study and its presentation.

1. Although the manuscript is well written, it should be improved, particularly the discussion. The reader would benefit from a graphical model integrating the findings. Using this model for navigation, the discussion should be rewritten.

We have prepared a graphical summary that is now included in the manuscript as Figure 8. The discussion has been rewritten and we trust that its clarity is improved considerably.

2. Fig.2: Several information is not presented in a uniform format or remains ambiguous. In this respect the manuscript appears preliminary and deserves improvement.

Examples: Labeling in 2D "FC vs WT Ctrl".

Statistical significance is not uniformly shown, compare e.g. 2A, F and the other subfigures.

Different font size used.

Please carefully revise.

We apologize for these inaccuracies in labeling. We have now corrected all figures to unify labeling formats and indication of statistical significance.

3. Quantitative PCR experiments: Sorl1 mRNA levels were quantified relative to Hprt1 (co-cultivation with glioma) or beta2M (LPS or PMA treatment). Why were different genes used for normalization? How was their suitability as reference genes tested? According to the MIQE guidelines (Bustin et al. 2009) more than one reference gene should be used for each gene of interest.

The choice of reference genes was based on pilot experiments where we assessed the expression levels of several candidate genes under our experimental conditions. For the sake of transparency, we now include the results of these experiments as Figure S2 in the revised manuscript. We have also carefully re-evaluated all qPCR data. To be more

accurate, we decided to change the reference gene used for TNF α transcript levels after PMA stimulation of WT and SORLA-KO primary microglia (from β 2M to Hprt1; Figure 3C in the revised manuscript). Please note that this modification did not change the results of our analysis.

4. Colocalization (Fig. 3A), should be quantified e.g. by calculating Mander's co-localization coefficient. Based on the current images colocalization is difficult to determine.

Colocalization of TNF α with SorLA has now been quantified as Mander's co-localization coefficient. We have also provided a representative image of better quality. Representative results of these analyzes are shown in Fig 4A-B.

5. Co-IP (Fig. 3B), high protein amounts were used in the input, but only low amounts of SorLA precipitated. Co-IP of partial receptors (Fig 3D), results in a high background, notably delta-EGF/betaP is precipitated (3D), but EGF/betaP (3E) seems to mediate the interaction. Together, the presented CoIPs do not convincingly show domain specific interactions.

Actually, the current blots suggest that precipitation conditions were not stringent enough.

The authors indicate a "ratio of pull-down and input signals (PD/IN)" which was "calculated for each transfection variant". Although the given numbers support the authors conclusions, how these have been calculated is difficult to follow. In the methods section the authors mention Western blot quantification. Are the numbers, which are given without statistical analysis, the result of a single experiments? This is not indicated and a great concern.

Clear cut experiments demonstrating the proposed interaction are essential to draw the functional conclusions made in the manuscript.

The study would greatly benefit from a demonstration of direct interaction of SorLA or its specific subdomains and TNFalpha using other, preferably biophysical, methods such as MST or Octet.

We fully agree that MST would provide additional means of analysis of SorLA-TNF α interactions. However, such analysis will require a considerable additional workload given the need to establish expression and purification system for full length and truncated forms of the receptor as well as its ligand TNFalpha. We strongly feel that these additional studies significantly exceed the scope and time envisioned for revision of this manuscript.

In the original figure (Fig. 3D-E), we showed a ratio of pull-down to input signals (PD/IN, IP/IN in the revised manuscript) which was calculated for each transfection variant, specifically for the presented exemplary western blot. In the revised manuscript, we now summarize the results of several independent experiments as graphs. We trust that this edit significantly improves the clarity of our data. The IP/IN ratio was calculated to adjust for the differences in the expression levels of the mutants.

Taken together, we are convinced that the results obtained for SorLA mini-receptors are robust and clearly point to the EGF/ β P domain as the region mediating the interaction with TNF α . However, we found it more challenging to improve the quality of the results for deletion mutants of the receptor as modulating co-IP conditions (more stringent conditions suggested by the reviewer) resulted in the loss of SorLA-TNF α binding (Figure 2 for reviewers). In the revised manuscript, we now present these results along with the

quantification of several independent experiments on Figure EV4. We modified the text to highlight the fact that, in contrast to the experiments with mini-receptors, this approach using deletion mutants did not lead to conclusive results.

We would also consider removing the data obtained for the deletion mutants from the manuscript. They are more complex to understand and we feel that it could be beneficial to only focus on the results generated with the mini receptors that are clear and robust. We would be happy to know the reviewer's feedback on whether the data on deletion constructs should be removed from the manuscript.

6. For colocalization analysis (Fig. 3G) Mander's co-localization coefficient M2 is shown. Please present also M1.

Both tM1 and tM2 are now shown (Figure 4G in the revised manuscript).

7. Fig. 3G, typo: ttM2 must be tM2

This typo has been corrected (now Figure 4G).

8. Figure legends:

Figure 1C: please indicate as in the results section that the 5 clusters with lowest and highest SorL1 levels are shown.

We now have indicated these facts in the figure legend.

9. The manuscript could be shortened by providing large parts of figure 1 and 5 as supplemental information.

We understand the reviewer's point of view. However, as the other two reviewers asked for additional bioinformatical analyses, we strongly feel that the data on scRNAseq should not be moved to the supplemental information section. In fact, more data were generated as part of the revision that are now presented in Figures 1, 2, EV1, EV2 and S1.

Regarding Figure 5 (Figure 6 in the revised manuscript), we wish to refrain from moving its parts to the supplement, as we believe that it contains important data obtained in the *in vivo* model, that are necessary to present the broader view on the glioma-related phenotypes in WT and SorLA-KO mice.

Referee #3:

In this study, Kaminska and colleagues have explored the role of SorLA in establishing a glioma-supportive brain microenvironment by unraveling its mechanism of action. I believe this study could make a valuable contribution to the field with some additional analyses as follows:

1- The authors utilized data generated by Abdelfattah et al. in 2022 to investigate the gene expression levels of SORL1. While this study includes the expression signature of glioma cells across a vast number of cells, it would be beneficial for the authors to compare the expression signature of their glioma-associated microglia (GAMs) with the expression signature of GAMs identified in other datasets, which are also cited by the authors in the paper (Sankowski et al., 2019). This comparison would help understand whether the GMA signature is consistent across multiple studies.

To address this comment, we run pilot analyses on several available datasets from human GBM (listed in Table S2). Two of them (Sankowski et al., *Nat Neurosci.* 2019, PMID: 31740814; Pombo-Antunes et al. *Nat Neurosci.* 2021, PMID: 33782623) were generated using CD45+ -sorted cells, so the population of cells analyzed was not similar to the one in Abdelfattah et al. (*Nat Commun.* 2023, PMID: 35140215) and could not be directly compared. In another study, single-nuclei, and not single-cell, approaches were used (Wang et al. *Nat Cancer* 2022, PMID: 36539501). Finally, Chen et al. (*Genome Med.* 2021, PMID: 34011400) used a combination of various datasets and newly generated data. Given these complications, we decided to repeat our key analyses on the dataset generated by Neftel et al. (*Cell* 2019, PMID: 31327527). We have to point out that the dataset published by Abdelfattah et al. contains about 10x more cells than the one published by Neftel and colleagues. Still, major points regarding (1) GAMs marker genes, (2) high *SORL1* expression in GAMs, (3) spectrum/variability of *SORL1* expression levels within GAMs population, and (4) marker genes of GAMs clusters with highest and lowest *SORL1* expression, were recapitulated in this analyses using new data (Neftel et al.). These new results are now presented in Figures EV1, S1, Table S8 and in respective Data Files.

2- Although the primary focus of the study is on the specific gene SORL1, performing functional analysis of GAMs using gene ontology tools could provide a more detailed understanding of the functional states of the microglia clusters.

Following the reviewer's suggestion, we performed GO analyzes of the GAMs clusters implementing commonly used approaches. Specifically, we performed Pathway Enrichment Analysis with clusterProfiler and Gene Set Enrichment Analysis of Gene Ontology, KEGG. However, these analyzes failed to generate any meaningful information on the functional properties of the GAMs clusters due to lack of statistical significance. We assume that the GAMs clusters do not show massive variability in gene expression, as these cells are not globally dissimilar – in fact, they have been pre-selected based on the similarities in expression of GAMs marker genes.

Thus, to provide a more detailed understanding of the functional states of GAMs with high and low *SORL1* expression, we used an alternative tool, based on Natural Language

Processing (NLP) of genes descriptions. Each gene description was gathered from various molecular databases provided by BiomaRt (Durinck et al., *Nat Protoc.* 2009, PMID: 19617889) such as: NCBI, Gene Ontology, KEGG, Reactome, WikiPathways, Biocarta; and later genes were clustered and sets of key words retrieved to uniquely describe each cluster. To perform this operation each gene was represented as the text document (combined from all available descriptions), and later, to calculate TF-IDF (term frequency–inverse document frequency), as a bag of words/terms. When each gene is represented by a TF-IDF vector it is possible to calculate cosine similarity between the documents and a hierarchical clustering algorithm can be applied based on these similarities. Finally, discovering a set of unique keywords that characterize each cluster provides a good functional biological overview of input genes and groups that should be studied more closely. This analysis indeed revealed functionally linked genes that we missed with our manual approach that was included in the manuscript. We incorporated these new results in the revised manuscript (Fig. 1C and Data Set 3).

3- To gain a better insight into the microenvironmental state of the tumor cells, the authors could conduct cell-cell interaction analysis using appropriate tools like CellChat or NichNet on the single-cell RNA-seq datasets. These analyses would shed light on the secretory profiles of the cells and, importantly, provide evidence regarding the receptors on the target cells, as the authors have claimed that the "presence of the receptor in host cells is critical for establishing a tumor-promoting microenvironment."

We would like to thank the reviewer for this suggestion. We performed CellChat analysis, which allowed us to obtain interesting and novel results concerning the properties of distinct cell populations and their interactions. The results of these analyses are now shown in the new Figures 2 and EV2. Related data required for cell populations annotation as given in Tables S5, S6, S7.

Of note, we realized that our statement that 'presence of the receptor in host cells is critical for establishing a tumor-promoting microenvironment' might have been misleading. What we meant by the 'receptor' was specifically SorLA. We now rephrased this statement.

.....

In addition to addressing reviewers' comments, we have introduced other improvements to the manuscript.

While re-evaluating our western blot data, we have realized that the signal detected for MLKL (Figure 6A in the previous version of the manuscript) cannot be fully trusted. In brief, we noted that signal coming from the primary antibodies produced in mouse in western blot analysis of our glioma samples might be obscured by detection of endogenous IgGs present in these samples. Thus, we decided to remove this western blot from our manuscript.

To further support the functional link between TNF α secretion and necroptosis of glioma cells, we now added the results of new experiments (Figure 7F). In these experiments, we treated GL261 glioma cells cultured in vitro with TNF α and we analyzed the levels of necroptosis marker pRIP1.

We also corrected editorial mistakes throughout the manuscript and figures.

Dear Dr. Malik,

Thank you for the submission of your revised manuscript to our editorial offices. I have now received the reports from the two referees that I asked to re-evaluate your study, you will find below. As you will see, the referees now fully supports the publication of the study in EMBO reports. Referee #2 has some further suggestions, I ask you to address in a final revised manuscript.

I would also suggest to include Fig. 2 for referees in the Appendix and to mention it in the manuscript text.

- Please name the datasets and call these out according to the nomenclature Dataset EVx (e.g. "Data Set 1" should be "Dataset EV1" etc.).
- Please add a title and a legend on the first TAB of each dataset excel file.
- Please name the Appendix items 'Appendix Figure Sx' and 'Appendix Table Sx' in the Appendix and use these names for their callouts. There is a "Table S8" mentioned in the text. Please correct this to "Appendix Table S8". Please carefully check that all items are correctly called out.
- Please make sure that the number "n" for how many independent experiments were performed, their nature (biological versus technical replicates), the bars and error bars (e.g. SEM, SD) and the test used to calculate p-values is indicated in the respective figure legends (for main, EV and Appendix figures) of the final revised manuscript. Please also check that all the p-values are explained in the legend, and that these fit to those shown in the figure. Please provide statistical testing where applicable. Please avoid the phrase 'independent experiment', but clearly state if these were biological or technical replicates. Please also indicate (e.g. with n.s.) if testing was performed, but the differences are not significant. In case n=2, please show the data as separate datapoints without error bars and statistics. See also:

<http://www.embopress.org/page/journal/14693178/authorguide#statisticalanalysis>

If $n < 5$, please show single datapoints for diagrams. It seems, that for some diagrams n.s. is missing or that they show no or only partial statistics (see e.g. 4B, 5B, EV4C, S2 and S5). Please check. Moreover:

- Please note that figure EV 4c does not contain any statistical information, kindly rectify the statistical test related information in the figure legend appropriately.
- Please note that the error bars are not defined in the legend of figure 7f.
- Please note that the scale bar needs to be defined for figure 1a.
- Please note that the white arrowheads are not defined in the legend of figure 1a. This needs to be rectified.
- Please add to each legend a 'Data Information' section explaining the statistics used or providing information regarding replicates and scales.

- Please remove the reagents and tools table from the main manuscript text file. I have attached templates for that in word or excel format. Please upload the filled in table to the manuscript tracking system as 'Reagent Table' file. Please also adjust any callouts to this table. The example linked below shows how the table will display in the published article and includes examples of the type of information that should be provided for the different categories of reagents and tools. Please list your reagents/tools using the categories provided in the template and do not add additional subheadings to the table. Reagents/tools that do not fit in any of the specific categories can be listed under "Other":

https://www.embopress.org/pb%2Dassets/embo-site/msb_177951_sample_FINAL.pdf

- Please make sure that all the funding information is also entered into the online submission system and that it is complete and similar to the one in the acknowledgement section of the manuscript text file. Presently, a grant (?) 'contract No 2022/WK/05' is only mentioned in the acknowledgements.

- We could not find the source data image for Fig. 1A at Biolineage (S-BSST1327). Please check.

In addition, I would need from you:

- a short, two-sentence summary of the manuscript (not more than 35 words).
- two to four short (!) bullet points highlighting the key findings of your study (two lines each).
- a schematic summary figure as separate file that provides a sketch of the major findings (not a data image) in jpeg or tiff format (with the exact width of 550 pixels and a height of not more than 400 pixels) that can be used as a visual synopsis on our

website.

Best,

Referee #1:

The authors have addressed all my comments/corrections.

Referee #2:

The authors have addressed all concerns I raised and significantly improved the manuscript. The Co-IP analysis is only partially improved. Precipitation of full-length SorLA and of extracellular domains of SorLA by GFP-TNFalpha is convincingly demonstrated. The authors now include additional independent experiments in their analysis that support their conclusions (Fig. 4F). Together, these experiments demonstrate interaction of TNFalpha and SorLA and that it is highly likely that SorLA binds TNFalpha predominantly through its EGF-type repeats and the beta-propeller.

The additional Co-IP experiment using SorLA deletion constructs could not be further improved by altering the precipitation conditions. I agree that purification of SorLA or parts of SorLA would likely exceed revision time and would not significantly improve the outcome of the study.

Although the Co-IP shown in EV4 is not entirely conclusive, I suggest to include the figure as expanded view content. Assuming that the domain borders in the deletion constructs were chosen correctly, the experiment does not rule out additional binding sites outside the EGF-type repeats and beta-propeller for TNFalpha in SorLA. This could be mentioned in the text.

All editorial and formatting issues were resolved by the authors.

Dr. Anna Malik
Faculty of Biology, University of Warsaw
Poland

Dear Dr. Malik,

I am very pleased to accept your manuscript for publication in the next available issue of EMBO reports. Thank you for your contribution to our journal.

Yours sincerely,
